# Long-term consumption of hydrogen-rich water provides hepatoprotection by improving mitochondrial biology and quality control in chronically stressed mice

Qi He[1,2], Xiang Lan[1,2], Mengyuan Ding[1,2], Na Zhang[3]*

1 College of Veterinary Medicine, Northeast Agricultural University, Harbin, China, 2 Heilongjiang Provincial Key Laboratory of Pathogenic Mechanism for Animal Disease and Comparative Medicine, Northeast Agricultural University, Harbin, China, 3 The Key Laboratory of Dairy Science of Education Ministry, Northeast Agricultural University, Harbin, China

☯ These authors contributed equally to this work.
* nazhang1981@126.com

## Abstract

### Background

Chronic stress has emerged as a prevalent facet of contemporary existence, significantly jeopardizing overall bodily health. The liver, a pivotal organ responsible for metabolic equilibrium, is particularly vulnerable to its adverse effects. This study delves into the hepatoprotective properties of extended consumption of HRW in mice subjected to chronic stress.

### Methods

Mice subjected to chronic stress via CUMS and HRW administration for seven months underwent liver pathological examination. Key liver function indicators (AST, ALT), oxidative stress markers (SOD, CAT, GSH), and markers related to lipid peroxidation and ferroptosis (MDA, Fe) were measured using standard kits. ELISA determined corticosterone and 4-HNE levels. Immunofluorescence evaluated ROS, Nrf2, and apoptosis in liver tissues. Western blotting analyzed markers for ferroptosis (GPX4, SLC7A11, HO-1, Nrf2), apoptosis (Bax, Bcl-2, Cytc, Caspase-3, Caspase-8), mitochondrial biogenesis (Nrf1, PGC-1α, Tfam), and quality control (Drp1, Fis1, Mfn1, Mfn2, OPA1, PINK1, Parkin, LC3 I/II).

### Results

The findings indicate a noteworthy improvement in liver health among mice exposed to HRW, as evidenced by histological analysis. Furthermore, the consumption of HRW exhibited hepatoprotection, as evidenced by the normalization of AST and ALT levels. Mechanistically, our results indicate that HRW elevates the levels of SOD, CAT, and GSH, while effectively clearing ROS within mitochondria. It was observed led to a regulation in the expression of mitochondrial quality control proteins, consequently improving mitochondrial biogenesis (Nrf1, PGC-1α, Tfam), and increasing ATP production. Furthermore, HRW

**Data Availability Statement:** All relevant data are within the article and its Supporting Information files.

**Funding:** This work was supported by the National Natural Science Foundation of China (Grant No. 32473107) and the Science and Technology Plan Project of Inner Mongolia Autonomous Region (Grant No. 2022YFSH0052).

**Competing interests:** The authors have declared that no competing interests exist.

decreased Cytc, Bax, Caspase-3, and Caspase-8 levels, and increasing the expression of Bcl-2. Additionally, HRW reduced MDA and 4-HNE levels, alleviating ferroptosis through the Nrf2/HO-1 pathway, and upregulating the expression of GPX4 and SLC7A11. By mitigating hepatocyte death through the aforementioned mechanisms, HRW fulfills its crucial role in safeguarding liver health.

## Conclusions

This study reveals that long-term hydrogen-rich water (HRW) consumption provides significant hepatoprotection in mice under chronic stress. HRW normalizes liver enzyme levels, enhances antioxidant capacity, and reduces lipid peroxidation and ferroptosis. It improves mitochondrial biogenesis, function, and ATP production, and attenuates apoptosis by modulating related proteins. Behavioral tests show HRW alleviates stress-induced anxiety and enhances exploratory behavior. These findings suggest HRW is a promising non-invasive intervention for preventing and treating stress-related liver disorders by targeting oxidative stress and mitochondrial dysfunction.

## Introduction

Stress, in its fundamental form, is the body's innate response to perceived threats or challenges [1]. It initiates a series of physiological and psychological reactions intended to help the body deal with the situation. Stress can be categorized based on the duration of the stimulus, distinguishing between acute stress and chronic stress [2]. Acute stress is typically beneficial to the body [3], while chronic stress is detrimental both physiologically and psychologically. Chronic stress occurs when these reactions persist over an extended period, often without any direct threat. Various sources, including social, psychological, and physiological pressures, can induce chronic stress [4]. Chronic stress significantly impacts overall physical health, elevating the risk of numerous chronic diseases such as cardiovascular disorders, diabetes, and immune system dysregulation [5, 6]. Therefore, investigating effective methods to alleviate chronic stress has become a vital research focus in modern medicine.

In this context, hydrogen gas has gained significant attention as an emerging medical gas [7]. Hydrogen-rich water (HRW), which contains a high concentration of hydrogen gas ($H_2$), offers a unique advantage due to its small molecular size and exceptional solubility. These properties allow it to easily permeate cell membranes and enter cells, exerting its biological effects. Recent research has shown that hydrogen-rich water can alleviate various liver oxidative stress-induced injuries. For instance, Sun et al. [8] found that hydrogen-rich saline inhibited harmful ROS in the livers of mice with experimental liver injury and reduced the activity of pro-apoptotic factors such as JNK and caspase-3, thereby exerting a protective effect against liver injury. In the study by Iketani et al. [9], pre-drinking hydrogen-rich water was found to reduce liver damage in mice with LPS-induced sepsis. Li et al. [10] found that Hydrogen-rich saline protected liver tissues from ischemia-reperfusion injury by alleviating liver tissue ERS. Lin et al. [11] discovered that hydrogen-rich water mitigated ethanol-induced mouse fatty liver through its antioxidant and anti-inflammatory effects. Moreover, numerous studies have reported that drinking hydrogen-rich water can increase hydrogen concentration in the liver, allowing hydrogen molecules to exert their effects within the liver [9].

This study aims to investigate the protective effects of long-term consumption of HRW on the livers of mice subjected to chronic stress. A comprehensive range of experimental

techniques, including behavioral observations, measurements of biochemical markers, histopathological analysis, Western blotting, and immunofluorescence, will be employed to systematically assess the impact of HRW on liver function and structure in mice experiencing chronic stress. The objective of these experiments is to uncover the mechanisms through which HRW influences mice under chronic stress, with a specific focus on the liver. This research endeavor aims to establish a scientific foundation for the clinical application of HRW. Through in-depth investigation, our goal is to provide theoretical support for the development of innovative stress resistance and liver protection strategies. Additionally, we aim to offer new insights into the clinical utilization of HRW, facilitate its broader application in the realm of chronic disease treatment, and contribute to the advancement of human and animal health.

## Materials & methods

### 2.1 Animals and experimental scheme

**2.1.1 Experimental animals grouping and experimental process.** KM (Kunming) mice of both genders (n = 48, male = 24; female = 24) were weighed 30±5g (Liaoning Changsheng Biotechnology Co., Ltd) and were housed with unrestricted access to food and provided ample space for free movement. The mice, all eight weeks old, were regularly checked (mental status, fur quality, fecal consistency and so on) for good health and did not have any underlying diseases before the start of the experiment. They were kept in a controlled environment at 20 ±1°C with a relative humidity of 55%, following a 12-hour light-dark cycle. Water was made available to the mice during specific intervals: 8:00–10:00 am, 1:00–3:00 pm, and 6:00–8:00 pm. The mice were acclimated to the environment for 7 months before the commencement of the experiments. Ethical approval for all animal experiments conducted in this study was obtained from the Ethics Committee for Experimental Animals of Northeast Agricultural University (NEAUEC2022 03 21).

Mice were randomly divided into 3 groups using simple randomization., 16 mice per group (n = 16, male = 8; female = 8): the HRW treatment group (HRW group), the model group (Model group), and the control group (C group). Chronic stress modeling began after seven months of HRW feeding. The Model and HRW groups were subjected to Chronic Unpredictable Mild Stress (CUMS) experiments involving a series of stressors: heat stress, cold stress, crowding, slanted cages, starvation, vibration, and tail clipping. The HRW treatment group received daily consumption of freshly prepared HRW, while the model group and control group received distilled water. In the context of behavioral experiments, it is essential to measure and statistically analyze various parameters: total movement distance, frequency of grid crossings, number of times standing, and duration of resting in mice. After behavioral tests, mice were anesthetized using isoflurane inhalation. Subsequently, their eyes were enucleated for blood collection, after which they were euthanized using cervical dislocation and liver tissues were extracted for further analysis. The livers were partitioned into two sections: one part was fixed in formalin for liver morphology analysis, and the remaining samples were rapidly frozen in liquid nitrogen and stored at -80°C for frozen section preparation and various index measurements.

**2.1.2 Preparation of HRW.** Hydrogen gas produced by the hydrogen generator (QL-500, Saikesaisi Hydrogen Energy Co, Shandong, China) is introduced into 300 mL of distilled water through aeration. After 15 minutes, it yields HRW with a concentration of 0.5 mg/ml. The concentration of hydrogen in the water decreases at an average rate of 0.1 mg/mL per hour at room temperature, as measured by a HRW detector. The HRW is prepared and utilized as required.

**2.1.3 Establishment of CUMS model.** After seven months of feeding, both the HRW group and the Model group underwent a 21-day CUMS protocol. This protocol consisted of three cycles of stress, each lasting seven days. Behavioral testing was conducted after the completion of each stress cycle. (1) heat stress: Place the mice in a constant temperature oven at 45°C for 1 hour. (2) cold stress: Place the mice in a 4°C refrigerator for 1 hour. (3) crowding: Combine four cages of mice into one cage, and maintain crowding for 24 hours. (4) slanted cages: Tilt the mice cage at a 45-degree angle and maintain for 24 hours. (5) starvation: Withhold food from the mice for 24 hours, but allow free access to water. (6) vibration: Place the mice on a shaker for 30 minutes. (7) tail pinching: Use a clip to clip the base of the mice tail, and maintain for 3 minutes.

## 2.2 Behavioral testing

The open field test was employed to assess the spatial exploration behavior and anxiety levels of the mice in an unfamiliar environment. Mice were given unrestricted movement within a black, uncovered square box measuring 100×100×40 cm, devoid of any light source, for a duration of 3 minutes. A digital camera positioned above the box recorded the test data. Prior to the open field test, mice were placed in the testing environment for 30 minutes to acclimate to their surroundings. Following the 30-minute adaptation period, each mouse was placed individually in the center of the box and allowed to move freely within the dark, quiet enclosure for 3 minutes while the camera recorded their movement trajectories, total distance traveled, immobility time, number of standing instances, and grid crossings. The box was cleaned after each mouse's activity to ensure the reliability of the results.

## 2.3 Corticosterone determination

Following the manufacturer's guidelines, 10% liver tissue samples were homogenized and analyzed using ELISA kits specific for corticosterone (Cort) (Jingmei Biotechnology, Jiangsu, China). Standard wells were set up, with each receiving 50μl of standard solution at varying concentrations. Sample wells received 10μl of the sample to be tested, followed by 40μl of sample diluent. Subsequently, 100μl of horseradish peroxidase (HRP)-conjugated detection antibody was added to each well, both in the standard and sample wells. Reaction wells were sealed with a sealing membrane and then incubated at 37°C for 60 minutes in a constant temperature incubator. After removal of the liquid, the plate was blotted dry with absorbent paper, and each well was filled with wash buffer, left to stand for 1 minute, and then flicked off. This washing process was repeated five times. Subsequently, each well received 50μl of substrate A and B, and the plate was incubated at 27°C in the dark for 15 minutes. Following this, 50μl of stop solution was added to each well, the absorbance of the samples (n = 5 per group) was measured at 450 nm using an Epoch microplate reader (BioTek, Winooski, VT, USA). Expression levels of 4-HNE and Cort were determined utilizing standard curves.

## 2.4 Biomarkers of liver toxicity

A 10% liver tissue homogenate was prepared by homogenizing the liver tissue with a ratio of 9 parts normal saline to 1 part liver tissue. The homogenate was then centrifuged at 2500 rpm for 10 minutes, and the resulting supernatant was collected. Alanine transaminase (ALT) and, aspartate transaminase (AST) test kit used (C009-2-1, C010-2-1), 20μl of pre-warmed substrate solution was added to both the test wells and the control wells. Subsequently, 5μl of the sample to be tested was added to the test wells, while no sample was added to the control wells. Following a 30-minute incubation at 37°C, 20μl of 2,4-dinitrophenylhydrazine solution was added to both the test and control wells. Additionally, 20μl of the sample to be tested was added to the

control wells, while no sample was added to the test wells. The plate was then incubated at 37°C for 20 minutes. Following this, 200μl of 0.4 mol/L sodium hydroxide solution was added to both the test and control wells. The plate was gently mixed and left at room temperature for 15 minutes. The optical density (OD) of each well was measured at 510nm using a microplate reader. The absolute OD value (OD of the test well minus OD of the control well) was used to construct a standard curve. The activity of ALT/AST was determined accordingly (n = 5 per group).

## 2.5 Liver enzymatic antioxidant activities

Superoxide dismutase test kit (SOD, A007-1-1, Jiancheng, Nanjing, China) is used according to the instructions. In the control and control blank wells, 20 μl of distilled water was added. Likewise, 20 μl of tissue homogenate was added to the test and test blank wells. Subsequently, 20 μl of enzyme working solution was added to the control and test wells, while 20 μl of enzyme dilution solution was added to the control blank and test blank wells. Following this, 200 μl of substrate application solution was added to the control, control blank, test, and test blank wells. After thorough mixing, the plates were incubated at 37°C for 20 minutes, and the OD values at 450 nm were determined using an enzyme marker. The SOD inhibition rate was calculated according to the formula: *Cpr: Protein concentration to be tested* (n = 5 per group).

$$SOD\ inhibition\ rate = \frac{[(OD_{control} - OD_{control\ blank}) - (OD_{test} - OD_{test\ blank})]}{(OD_{control} - OD_{control\ blank})} \times 100\%$$

$$SOD\ activity = \frac{SOD\ inhibition\ rate \times 50\%}{Cpr}$$

Catalase (CAT, A007-1-1, Jiancheng, Nanjing, China) test kit is used according to the instructions, 0.05mL of tissue homogenate was added to the test tube, while the control tube received no tissue homogenate. Subsequently, pre-warmed reagent one (1.0mL) and reagent two (0.1mL) were added to both the control and test tubes immediately after mixing, and the timer was started. After precisely 60 seconds at 37°C, reagent three (1.0mL) and reagent four (0.1mL) were added to both the control and test tubes. Additionally, 0.05mL of tissue homogenate was added to the control tube, while the test tube remained unchanged. After thorough mixing, the absorbance of each tube (n = 5 per group) was measured at a wavelength of 405nm. CAT activity was then calculated using the formula: *T: reaction time, 60s.*

$$CAT\ activity = (OD_{control} - OD_{test}) \times 271 \div V_{sample} \div T \div Cpr$$

Glutathione (GSH, A006-2-1, Jiancheng, Nanjing, China) test kit is used according to the instructions., In the blank wells, 100μl of reagent one was added, while in the standard wells, 100μl of 20μmol/L GSH standard solution was added. In the test wells, 100μl of tissue homogenate was added. Subsequently, 25μl of reagent three and 100μl of reagent two were added to the blank, standard, and test wells. The plate was gently shaken and left to stand for 5 minutes. The absorbance of each well (n = 5 per group) was measured at 405nm. GSH content was calculated using the formula:

$$GSH\ content = \frac{(OD_{test} - OD_{blank})}{(OD_{standard} - OD_{blank})} \times c_{standard} \times 5 \times 10$$

## 2.6 Liver lipid peroxidation detection

Malondialdehyde (MDA, A003-1, Jiancheng, Nanjing, China) test kit used according to the instructions, In the blank tubes, 100μl of anhydrous ethanol was added, while in the standard

tubes, 100μl of 10nmol/ml standard solution was added. The test and control tubes received 100μl of tissue homogenate. Subsequently, 100μl of reagent one was added to the blank, standard, test, and control tubes, followed by thorough mixing. Then, 3ml of reagent two was added to the blank, standard, test, and control tubes, and 1ml of reagent three was added to the blank, standard, and test tubes, with 1ml of 50% ice-cold acetic acid added to the control tubes. Cover the centrifuge tubes with caps, puncture a small hole in the cap using a needle, vortex mix thoroughly, immerse in a 95°C water bath for 40 minutes, remove, and cool under running water, then centrifuge at 3500–4000 rpm for 10 minutes. Collect the supernatant and measure the absorbance values of each tube (n = 5 per group) at 532 nm. MDA content was calculated using the formula:

$$MDA\ content = \frac{(OD_{test} - OD_{control})}{(OD_{standard} - OD_{blank})} \times c_{standard} \div Cpr$$

The ELISA detection method for 4-hydroxynonenal (4-HNE, Jingmei Biotechnology, Jiangsu, China) is the same as the corticosterone detection (Method 2.3).

## 2.7 ATP content detection

Following the instructions provided with the ATP content testing kit (A095-1-1, Jiancheng, Nanjing, China), 30 μL of 1 mmol/L standard solution was added to the blank tubes and standard tubes, while 30 μL of sample was added to the test tubes and control tubes. Subsequently, substrate solution one (100 μL) and substrate solution two (200 μL) were added to the blank tubes, standard tubes, test tubes, and control tubes. 30 μL of enhancer was added to the standard tubes and test tubes. After thorough mixing, the tubes were incubated in a 37°C water bath for 30 minutes. Then, 50 μL of precipitant was added to the blank tubes, standard tubes, test tubes, and control tubes. After thorough mixing, the tubes were centrifuged at 4000 rpm for 5 minutes, and 300 μl of supernatant was collected for measurement. Subsequently, 500 μL of color developing solution was added to all tubes, mixed thoroughly, and allowed to stand at room temperature for 2 minutes. Then, 500 μL of stop solution was added to all tubes, mixed thoroughly, and allowed to stand for 5 minutes. The absorbance of each tube was measured at 636 nm. ATP content was calculated using the formula: *N*: *the dilution factor of the sample before measurement.*

$$ATP\ content = \frac{(OD_{test} - OD_{control})}{(OD_{standard} - OD_{blank})} \times c_{standard} \times N \div Cpr$$

## 2.8 Tissue iron content detection

According to the instructions of the tissue iron test kit, the iron content (Fe, A039-2-1, Jiancheng, Nanjing, China) in liver tissue was determined. 500μl of double-distilled water was added to the blank wells, 500μl of iron standard application solution with a concentration of 2mg/L was added to the standard wells, and 500μl of the test sample was added to the test wells. Subsequently, 1500μl of iron chromogenic reagent was added to all wells, mixed thoroughly, heated in a boiling water bath for 5 minutes, cooled under running water, centrifuged at 3500 rpm for ten minutes, and 1000μl of supernatant was collected. The absorbance OD values of each well were measured at a wavelength of 520nm.

$$iron\ content = \frac{(OD_{test} - OD_{blank})}{(OD_{standard} - OD_{blank})} \times c_{standard} \div Cpr$$

## 2.9 Histological examination

Liver tissues were fixed in 4% formaldehyde for more than 24 hours, then processed for routine paraffin section preparation and stained with hematoxylin and eosin (HE). Morphological changes in the liver tissues of each group of mice, such as degeneration, necrosis, inflammatory cell infiltration, and other pathological alterations, were observed under an optical microscope.

## 2.10 Immunofluorescence

For TUNEL and Nrf2 assays, tissue samples were deparaffinized in water and rinsed twice for 3 minutes each with distilled water. Subsequently, the tissue samples were treated with proteinase K, followed by PBS washing. The tissues were incubated with the TUNEL/Nrf2 reaction solution and stained with DAPI for nuclear labeling. Finally, the slides were sealed with an anti-fade solution, and images were captured using a fluorescence microscope.

The liver tissue was sliced into 2 mm thick sections, embedded in cryoembedding medium, and then cut into 10 μm thick sections using a cryostat. Subsequently, the sections were incubated in a dark chamber containing dihydroethidium for 30 minutes, followed by a PBS wash. The samples were observed and imaged using a fluorescence microscope.

## 2.11 Western blotting

Protein is extracted using conventional RIPA lysate and PMSF solution. Nuclear protein and cytoplasmic protein extraction kit (P0027; Beyotime) was used to separate and extract nuclear and cytoplasmic protein from liver tissue, and used to detect Nrf2 protein. Proteins were separated using different concentrations of SDS-PAGE (6%-15%). The separated proteins on the electrophoresis gel were transferred onto a nitrocellulose (NC) membrane and blocked with 5% BSA at room temperature for 2 hours. The membrane was then incubated with the appropriate primary antibodies overnight at 4°C, the antibodies used are listed in Table 1. Subsequently, the NC membrane was incubated with secondary antibodies (goat anti-rabbit IgG, dilution 1:10,000) for 1 hour, followed by washing the membrane with TBST buffer. Next, protein bands were visualized using an ECL chemiluminescence kit (Beyotime Biotechnology, Shanghai, China). Images were captured using the Tanon-5200 chemiluminescence imaging system (Tanon Science & Technology, Shanghai, China). The grayscale values for each band were quantified using ImageJ software (NIH, Bethesda, MD, USA). Protein expression was normalized to the corresponding internal control (β-actin).

## 2.12 Statistical analysis

Data analysis and graphing were performed using GraphPad Prism 8 (GraphPad Software, San Diego, USA). The results were presented as mean ± standard error of mean (mean ± SEM) for all experimental data. One-way analysis of variance (ANOVA) was utilized for significant differential analysis, and Tukey test are used after ANOVA. Differences with a p-value less than 0.05 were considered statistically significant.

## Results

### 3.1 Behavioral testing

The results of the open field test for mice are depicted in Fig 1. Mice in C group exhibited diverse activities characterized by unpredictable movement patterns. In contrast, mice in the Model group displayed limited activity, mostly confined to the periphery. When compared to

**Table 1. Antibody description.**

| Antibody name | Art.No. | Manufacturer | Dilution ratio |
|---|---|---|---|
| Nrf1 | bs-1342R | Bioss | 1:1000 |
| OPA1 | bs-11764R | Bioss | 1:1000 |
| Mfn1 | bs-0557R | Bioss | 1:1000 |
| Mfn2 | bs-2988R | Bioss | 1:1000 |
| Tfam | A13552 | ABclonal | 1:1000 |
| Fis1 | A19666 | ABclonal | 1:1000 |
| PGC-1α | WL02123 | Wanlei | 1:1000 |
| DRP1 | WL03028 | Wanlei | 1:1000 |
| GPX4 | WL0546 | Wanlei | 1:1000 |
| SLC7A11 | A13685 | ABclonal | 1:1000 |
| HO-1 | WL02400 | Wanlei | 1:1000 |
| Nrf2 | WL02135 | Wanlei | 1:1000 |
| Cytc | WL02410 | Wanlei | 1:1000 |
| Bax | WL01637 | Wanlei | 1:1000 |
| Bcl-2 | WL01556 | Wanlei | 1:1000 |
| Caspase-3 | WL02117 | Wanlei | 1:1000 |
| Caspase-8 | WL03426 | Wanlei | 1:1000 |
| PINK1 | WL04963 | Wanlei | 1:1000 |
| Parkin | WL02512 | Wanlei | 1:1000 |
| LC3 I/II | WL01506 | Wanlei | 1:1000 |

the Model group, mice in the HRW group demonstrated a wider activity range with irregular movement patterns, resembling those in the C group.

The statistical analysis of mouse behavior in the open field test yielded noteworthy differences among groups. Specifically, the Model group displayed a substantial reduction in total distance traveled, grid crossings, and standing counts in comparison to C group ($P < 0.05$). Conversely, the Model group exhibited a significant increase in immobility time ($P < 0.05$). In contrast, the HRW group demonstrated a significant rise in total distance traveled, grid crossings, and standing counts in contrast to the Model group ($P < 0.05$). Additionally, the HRW group displayed a marked decrease in immobility time compared to both the Model and C groups ($P < 0.05$). Notably, the HRW group also showed a significant decrease in total distance traveled compared to Group C ($P < 0.05$).

### 3.2 Corticosterone level

Fig 2 provides clear evidence that both the Model group and the HRW group displayed a significant increase in corticosterone levels in comparison to C Group ($P < 0.05$). Additionally, the HRW group exhibited a noteworthy elevation in corticosterone levels when compared to the Model group ($P < 0.05$).

### 3.3 Liver structure and functional index

As depicted in Fig 3, the histological examination revealed distinctive findings across the different groups. C group exhibited no apparent liver structural damage. In contrast, the Model group displayed multiple noticeable bleeding spots, partial vacuolization of liver cells, and disorganized, separated hepatic cords. Remarkably, these pathological changes were significantly ameliorated in the HRW group. Upon analyzing liver function markers, it was observed that both the Model group and the HRW group exhibited a substantial elevation in ALT and AST

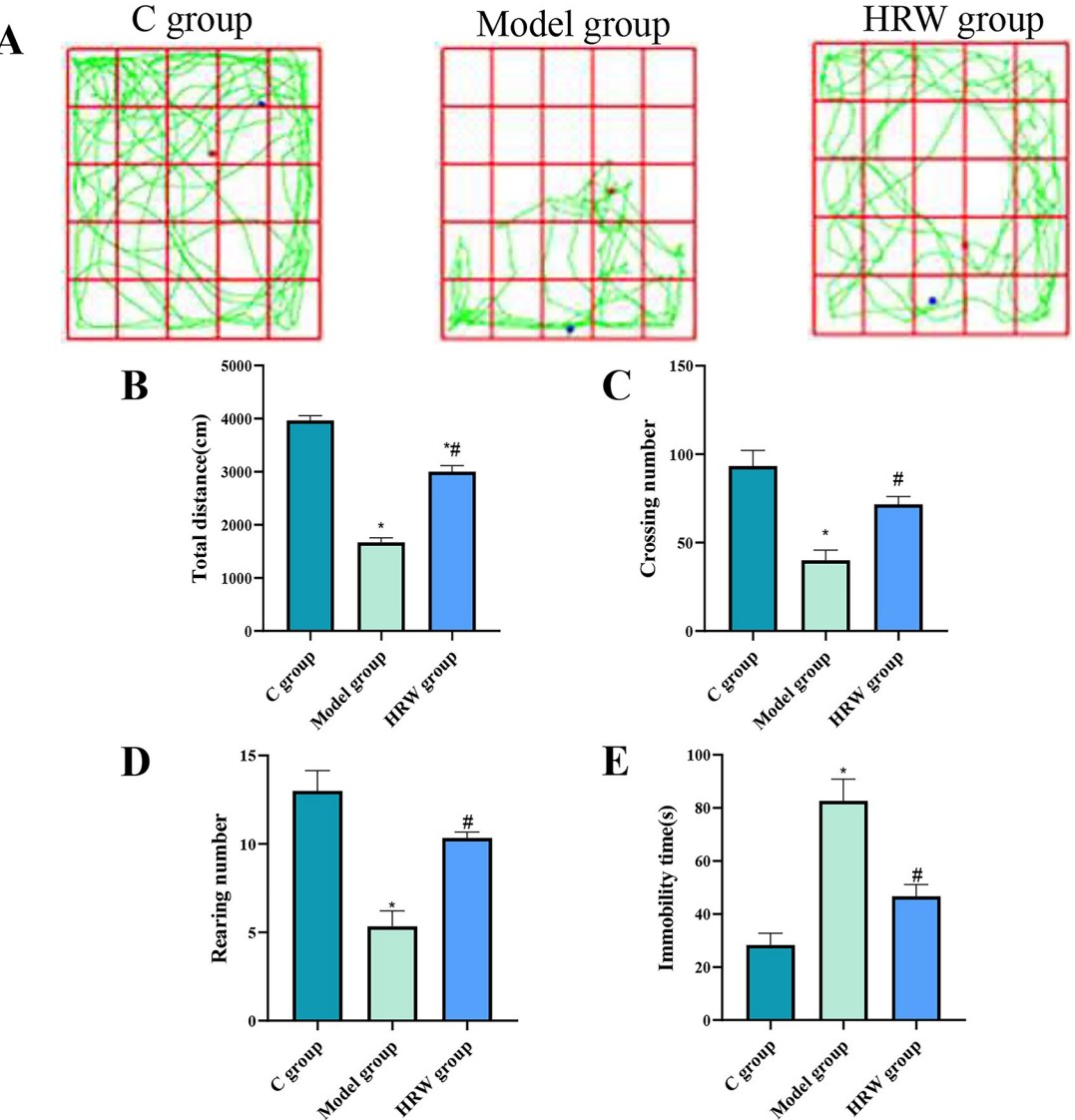

**Fig 1. Movement analysis of mice in open field experiments.** (A) After establishing a chronic stress model, the action tracks of C group, model group and HRW group. After establishing a chronic stress model, the assay of (B) the total distance. (C) Crossing number. (D) the rearing. (E) the immobility time in C group, model group and HRW group. Compared to C group, * *P<0.05*; Compared to Model group, *# P<0.05*.

levels compared to C group (P < 0.01). However, in comparison to the Model group, the HRW group demonstrated a significant reduction in ALT and AST content (P < 0.05).

## 3.4 Oxidative stress related factors

Fig 4 clearly illustrate that both the Model group and the HRW treatment group displayed a significant increase in ROS levels when compared to C group. However, it is noteworthy that the Model group exhibited a notably higher level of ROS within the liver when compared to the HRW group. In contrast, when compared to C group, both the Model group and the HRW group exhibited a significant decrease in SOD, CAT, and GSH levels (P < 0.05). Conversely, the HRW group displayed a substantial increase in SOD, CAT, and GSH levels within the liver when compared to the Model group (P < 0.05).

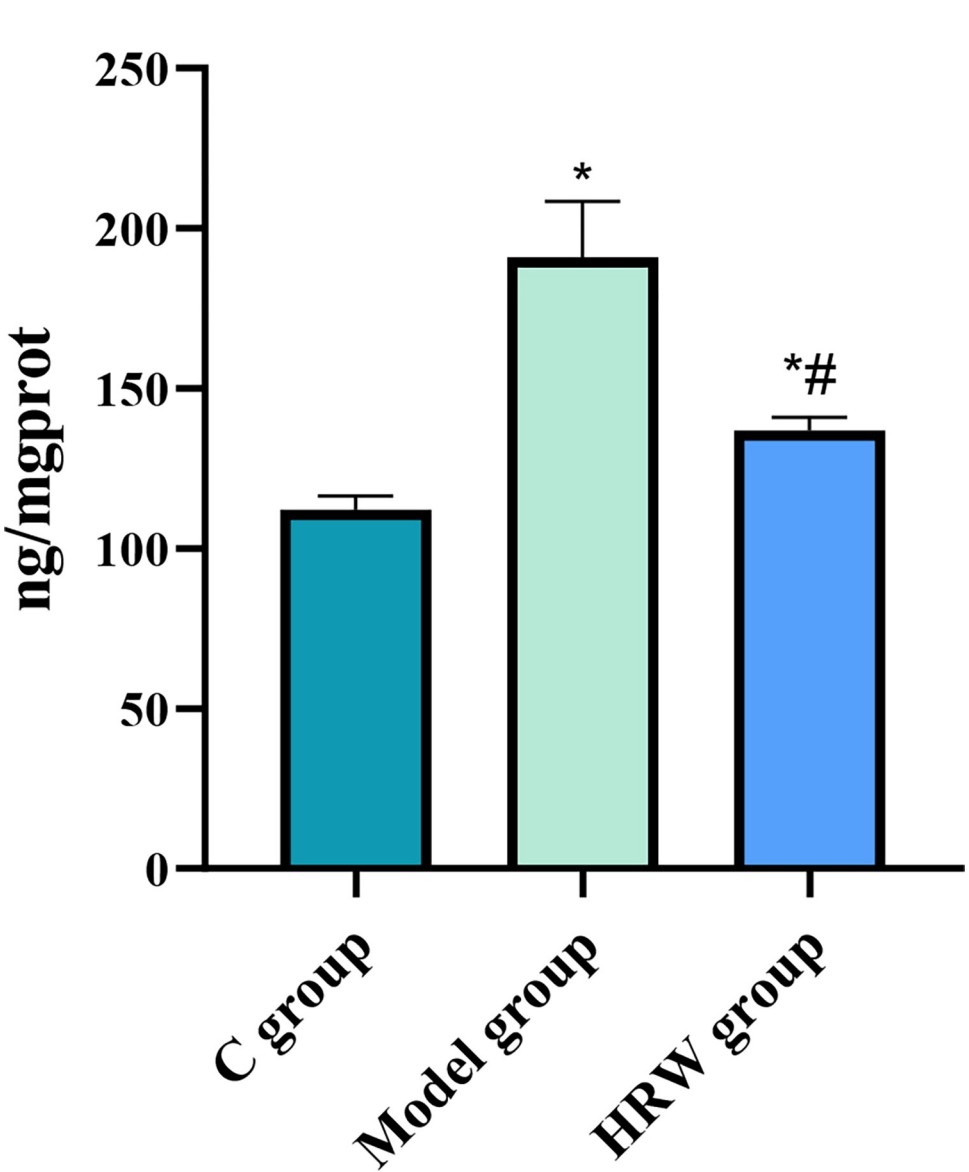

**Fig 2. Chronic stress mice drinking HRW decreased corticosterone levels.** Compared to C group, * *P<0.05*; Compared to Model group, # *P<0.05*.

### 3.5 Lipid peroxidation related factors

Fig 5 illustrates that the lipid peroxidation markers MDA and 4-HNE in C group were significantly lower than those in the Model group and HRW group (P < 0.05). Moreover, compared to the Model group, the HRW group exhibited a significant decrease in the levels of MDA and 4-HNE (P < 0.05).

### 3.6 Ferroptosis related factors

Based on the data presented in Fig 6, it is evident that both the Model group and the HRW group exhibited a significant increase in iron (Fe) content in comparison to C group

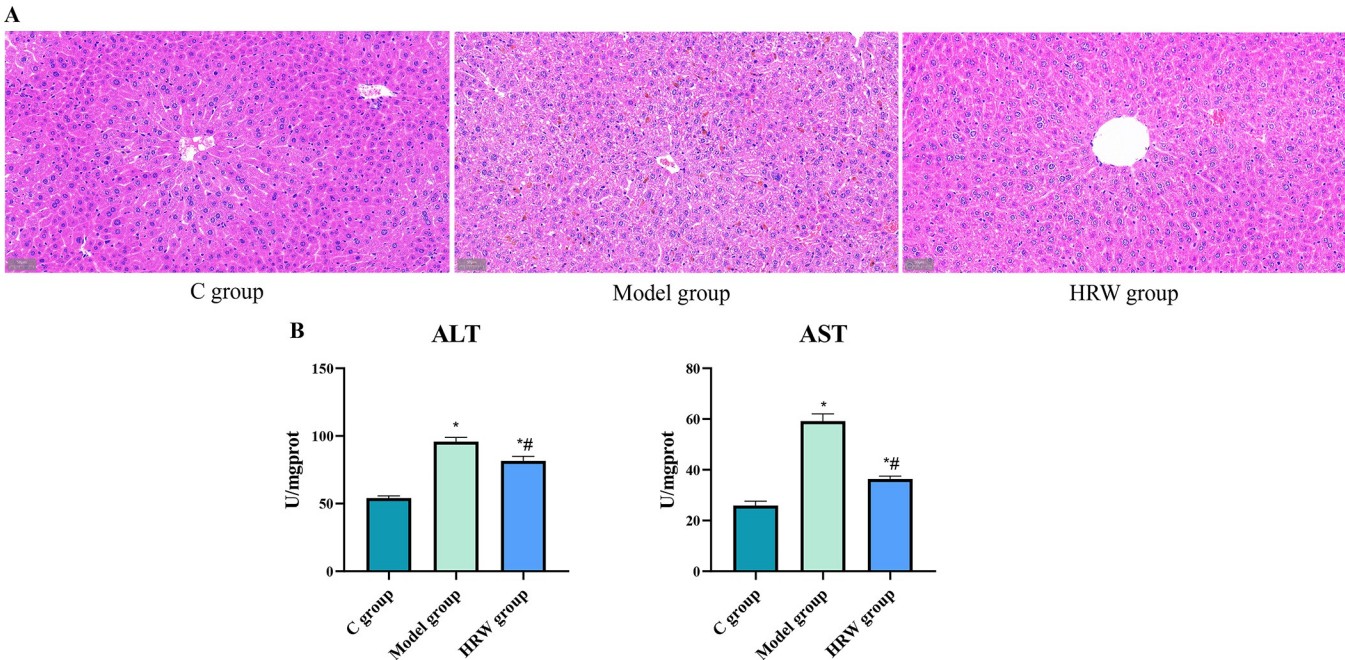

**Fig 3. The consumption of HRW by mice experiencing chronic stress resulted in a notable reduction in liver damage.** (A) After establishing a chronic stress model, pathological examination of C group, model group and HRW group. (B) After establishing a chronic stress model, the assay of ALT and AST. Compared to C group, * *P<0.05*; Compared to Model group, # *P<0.05*.

(P < 0.05). Notably, the HRW group displayed a significant decrease in Fe content when compared to the Model group (P < 0.05). Furthermore, when compared to C group, both the Model group and the HRW group showed a significant reduction in the protein expression levels of iron death-related markers GPX4 and SLC7A11 (P < 0.05). Additionally, the Model group exhibited significantly lower GPX4 and SLC7A11 protein expression levels in comparison to the HRW group (P < 0.05).

Upon analyzing proteins associated with the Nrf2/HO-1 pathway, it was observed that both the Model group and the HRW group exhibited significantly lower cytoplasmic Nrf2

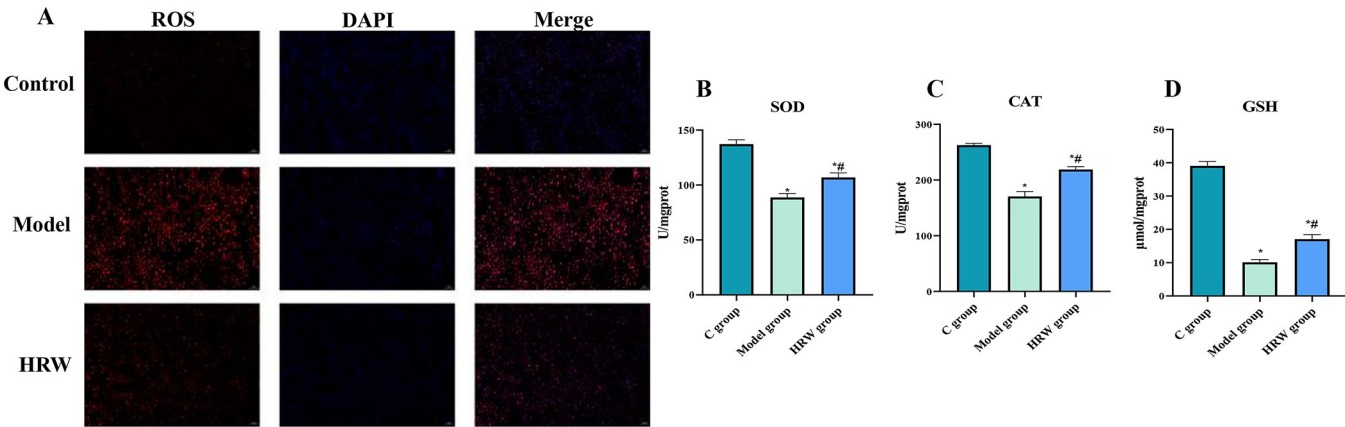

**Fig 4. Mice subjected to chronic stress and provided with HRW exhibited a significant reduction in oxidative stress levels.** (A) After establishing a chronic stress model, ROS immunofluorescence of C group, model group and HRW group. (B) After establishing a chronic stress model, the assay of antioxidant enzyme (B) SOD, (C) CAT, (D) GSH. Compared to C group, * *P<0.05*; Compared to Model group, # *P<0.05*.

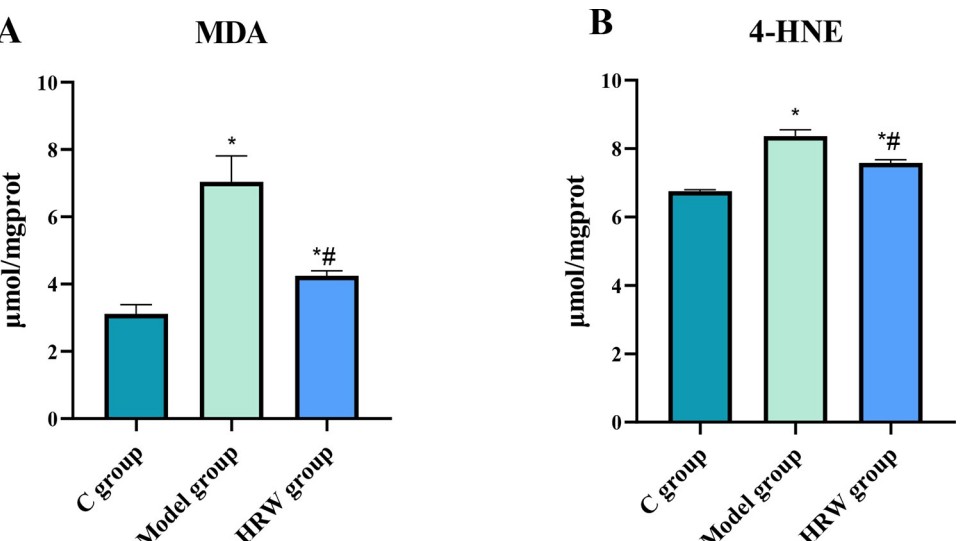

**Fig 5. Mice subjected to chronic stress and provided with HRW exhibited a significant reduction in lipid peroxidation levels.** After establishing a chronic stress model, the assay of antioxidant enzyme (A) MDA, (B) 4-HNE. Compared to C group, * *P<0.05*; Compared to Model group, # *P<0.05*.

expression levels compared to C group (P < 0.05). Furthermore, in comparison to the cytoplasmic Nrf2 protein expression in the Model group, the HRW group demonstrated a further decrease (P < 0.05). Conversely, concerning nuclear Nrf2 expression levels in comparison to C group, both the Model group and the HRW group displayed significantly higher levels (P < 0.05). Additionally, in contrast to the nuclear Nrf2 protein expression in the HRW group, the Model group displayed a significant decrease (P < 0.05). Furthermore, both the Model group and the HRW group exhibited a significant decrease in HO-1 protein expression levels when compared to C group (P < 0.05). Conversely, the HRW group showed a significant increase in HO-1 protein expression levels compared to the Model group (P < 0.05). This observation was further corroborated through immunofluorescence analysis of Nrf2, as depicted in Figs 3–6H.

### 3.7 Apoptosis related factors

The TUNEL fluorescence staining results in Fig 7 indicate a significant increase in apoptosis in both the Model group and the HRW group compared to the C group. Moreover, apoptosis was found to be reduced in the HRW group when compared to the Model group.

In the analysis of apoptosis-related protein expression, it was evident that both the Model group and the HRW group displayed a significant increase in the expression of pro-apoptotic factors, which included Bax, Cytc, Caspase-3, and Caspase-8, compared to the C group (P < 0.05). Notably, the HRW group exhibited a significant decrease in the expression of these pro-apoptotic factors when compared to the Model group (P < 0.05). Furthermore, in comparison to the C group, both the Model group and the HRW group demonstrated a significant decrease in the expression of the anti-apoptotic factor Bcl-2 (P < 0.05). Additionally, the expression of Bcl-2 was significantly lower in the Model group when compared to the HRW group (P < 0.05).

### 3.8 Mitochondrial biosynthesis related factors

As presented in Fig 8, both the Model group and the HRW group exhibited a significant decrease in ATP levels compared to the C group (P < 0.05). Furthermore, the HRW group

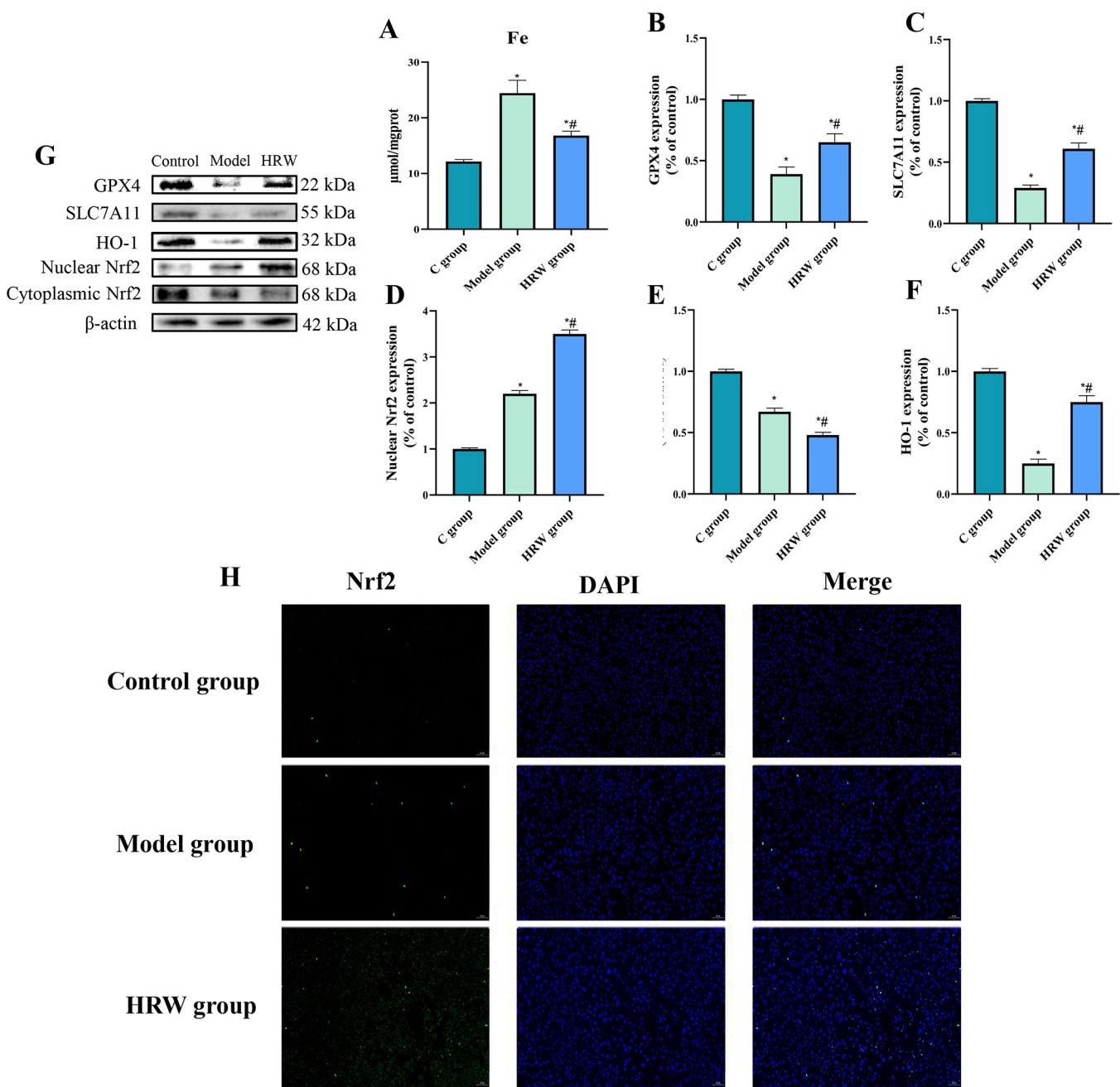

**Fig 6. Activation of Nrf2/HO-1 pathway by HRW alleviates ferroptosis in chronically stressed mice.** (A) After establishing a chronic stress model, the iron level of C group, model group and HRW group. After establishing a chronic stress model, analysis of the Western Blot (WB) for ferroptosis-related protein (B) GPX4, (C) SLC7A11, (D) Nuclear Nrf2, (E) Cytoplasmic Nrf2, (F) HO-1. (G) Western blot image. (H) Nrf2 immunofluorescence. Compared to C group, * $P<0.05$; Compared to Model group, # $P<0.05$.

displayed a significant increase in ATP levels when compared to the Model group (P < 0.05). In the analysis of factors associated with mitochondrial biogenesis, it was noted that both the Model group and the HRW group showed a significant reduction in the protein expression levels of PGC-1α, Nrf1, and Tfam in comparison to the C group (P < 0.05). Additionally, when compared to the Model group, the HRW group demonstrated a significant increase in the protein expression levels of PGC-1α, Nrf1, and Tfam (P < 0.05).

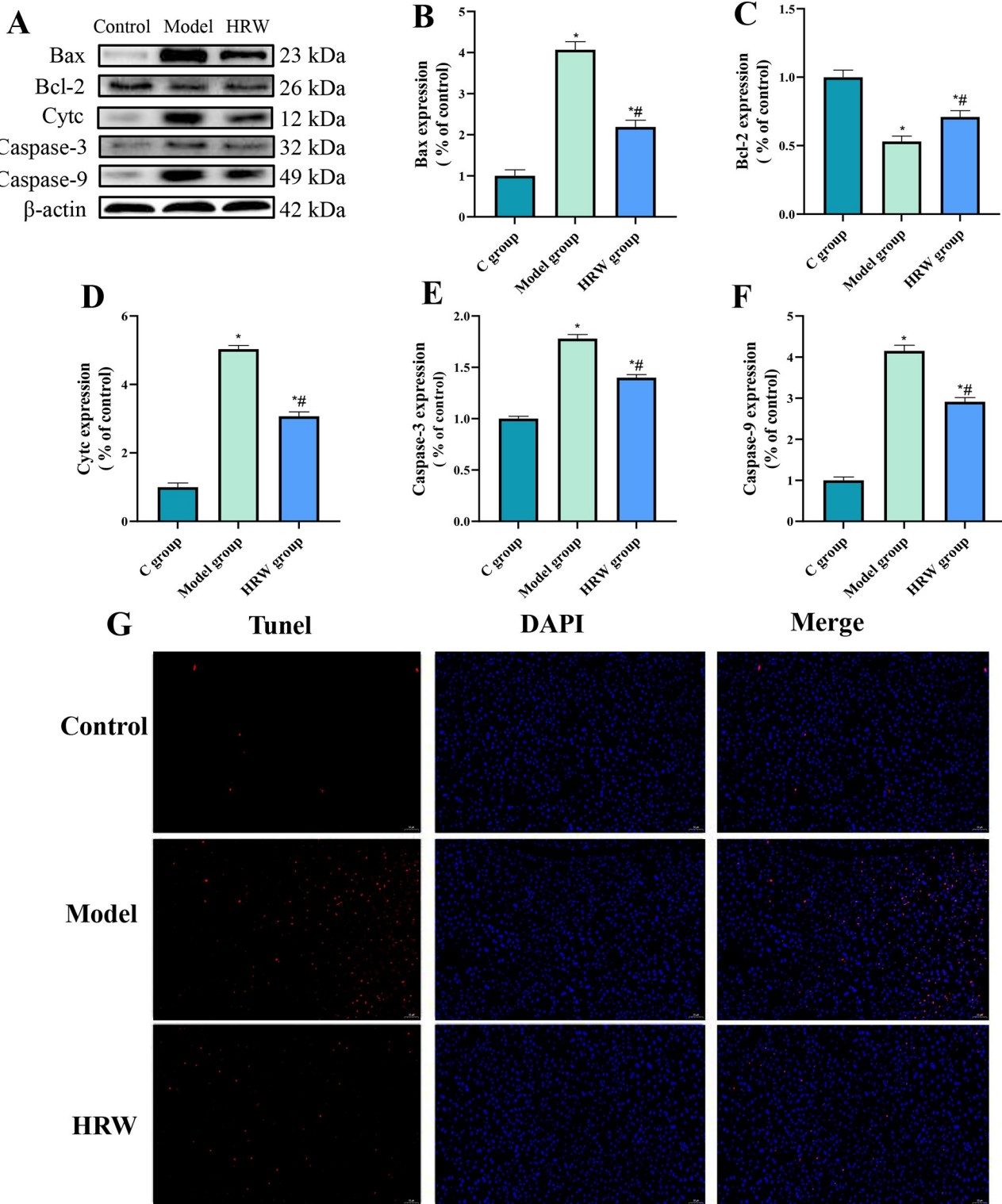

**Fig 7. HRW consumption reduces apoptosis in the liver of chronically stressed mice.** (A) Western blot image. After establishing a chronic stress model, analysis of the WB for ferroptosis-related protein (B) Bax, (C) Bcl-2, (D) Cytc, (E) Caspase-3, (F) Caspase-9. (G) Nrf2 immunofluorescence. Compared to C group, * *P<0.05*; Compared to Model group, # *P<0.05*.

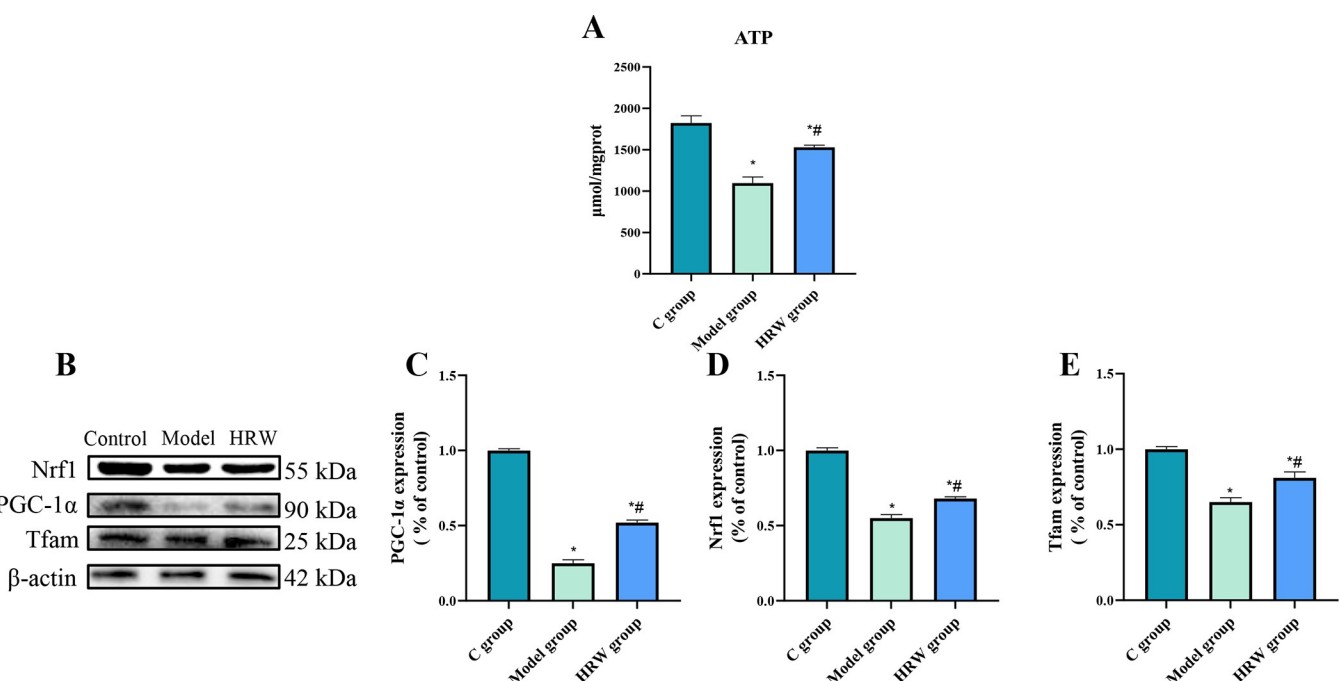

**Fig 8. HRW enhances mitochondrial function and biogenesis in chronically stressed mice.** (A) After establishing a chronic stress model, ATP level of C group, model group and HRW group. (B) Western blot image. After establishing a chronic stress model, analysis of the WB for ferroptosis-related protein (C) PGC-1α, (D) Nrf1, (E) Tfam. Compared to C group, * *P<0.05*; Compared to Model group, # *P<0.05*.

## 3.9 Mitochondrial quality control-regulated factors

Based on the data presented in Fig 9, it was observed that both the Model group and the HRW group exhibited a significant increase in the protein expression levels of mitochondrial fission factors, Drp1 and Fis1, compared to the C group ($P < 0.05$). Moreover, in comparison to the Model group, the HRW group displayed a significant decrease in the expression of Drp1 and Fis1 ($P < 0.05$). Regarding mitochondrial fusion factors, including Mfn1, Mfn2, and OPA1, both the Model group and the HRW group demonstrated a significant decrease in protein expression levels compared to the C group ($P < 0.05$). Notably, the HRW group exhibited a significant increase in the expression of Mfn1, Mfn2, and OPA1 when compared to the Model group ($P < 0.05$).

In the analysis of factors related to mitophagy, including PINK1, Parkin, and LC3I/II, it was noted that both the Model group and the HRW group exhibited a significant decrease in the expression levels of these proteins compared to the C group ($P < 0.05$). However, the HRW group displayed a significant increase in the expression levels of PINK1, Parkin, and LC3I/II when compared to the Model group ($P < 0.05$).

## Discussion

### 1. Establishment of a chronic stress model in mice

Our study successfully established a chronic stress model using the CUMS paradigm, as demonstrated by significant behavioral changes in the open field test and elevated corticosterone levels. Consistent with previous research, CUMS reliably induces behavioral and physiological changes in rodents that mimic human stress responses [12]. These findings strengthen the model's validity as a preclinical tool for exploring potential treatments for stress-related disorders.

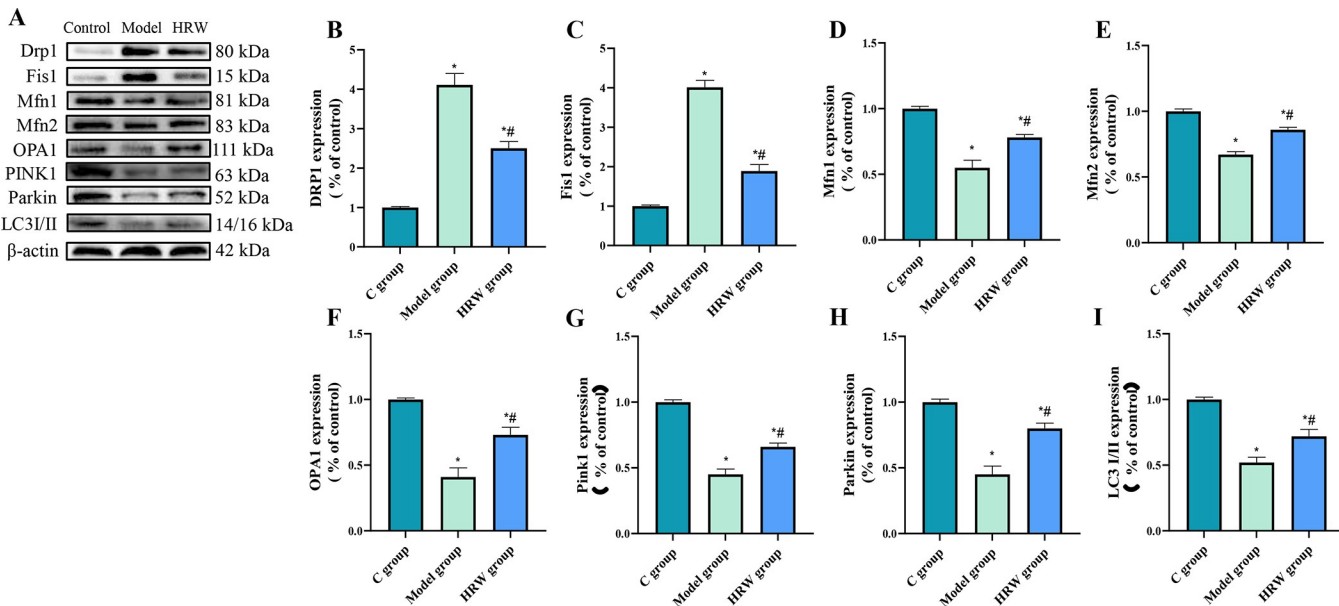

**Fig 9. HRW improves mitochondrial quality control in chronically stressed mice.** (A) Western blot image. After establishing a chronic stress model, analysis of the WB for ferroptosis-related protein (B) DRP1, (C) Fis1, (D) Mfn1, (E) Mfn2, (F) OPA1, (G) Pink1, (H) Parkin, (I) LC3 I/II. Compared to C group, * P<0.05; Compared to Model group, # P<0.05.

The reduction in corticosterone levels in HRW-treated mice suggests a moderating effect on the hypothalamic-pituitary-adrenal (HPA) axis, a core pathway implicated in stress responses [13]. Given that HPA axis dysregulation is also associated with anxiety and depression in humans, these results indicate HRW's potential as a therapeutic agent. However, it is important to recognize the challenges involved; while CUMS is effective in rodents, stress responses in humans are influenced by additional complex psychological and social factors not fully replicable in animal models. Further research is needed to determine HRW's efficacy and safety in human subjects and to explore optimal dosing strategies, potential side effects, and long-term impacts.

Notably, our findings align with previous studies showing that hydrogen-rich solutions possess anxiolytic and antidepressant effects in animal models [14]. This study contributes to the growing evidence that HRW might hold promise as a novel intervention for stress-related disorders, potentially offering an alternative approach that targets physiological stress pathways. The observed effect on corticosterone levels highlights a possible application of HRW in moderating stress without the need for traditional pharmacological treatments that may have more extensive side effects.

## 2. The effect of HRW on liver structure and function

This study clearly demonstrated the protective effects of HRW on liver structure and function through histological and biochemical analyses. Elevated ALT and AST levels are common indicators of liver damage [15], and our findings that HRW reduces these levels in chronically stressed mice align with previous studies demonstrating its hepatoprotective properties [16]. The ability of HRW to improve liver function under stress suggests its potential as a therapeutic agent for stress-induced hepatic damage [17]. Research has shown that hydrogen-rich solutions can protect liver tissue by enhancing antioxidant defenses and reducing inflammation [18]. Although our study demonstrates that HRW not only alleviates biochemical markers of

liver damage but also improves histological architecture, more comprehensive investigations are warranted to elucidate its underlying mechanisms. It is particularly important to assess the applicability of HRW in different animal models and clinical contexts, for example, while our results show the efficacy of HRW in mouse models, the application of these findings to humans requires consideration of physiological differences, dose optimization, and the safety of long-term use. Our findings highlight the potential of HRW in clinical applications, particularly in treating liver damage induced by chronic stress. Future clinical trials will be essential to validate the therapeutic effects of HRW, further advancing its application in liver disease management.

### 3. The effect of HRW on mitochondria in chronically stressed mice

Chronic stress is known to disrupt mitochondrial dynamics, increasing ROS production and impairing mitochondrial quality [19]. Our research demonstrated that HRW significantly improves mitochondrial function, as indicated by enhanced ATP levels and the regulation of mitochondrial biogenesis factors such as PGC-1α and Tfam. This supports previous findings where molecular hydrogen was shown to exert protective effects on mitochondria, potentially through its antioxidant properties [20]. Furthermore, the observed modulation of mitochondrial fusion and fission proteins, with increased Mfn2 and decreased Drp1 expression, suggests that HRW enhances mitochondrial dynamics, promoting fusion and reducing excessive fission, which is crucial for maintaining mitochondrial integrity under stress condition [21]. This is in line with studies showing that improved mitochondrial fusion can counteract stress-induced mitochondrial damage [22]. Additionally, the activation of mitophagy observed in our study supports the role of HRW in maintaining mitochondrial quality control, as mitophagy is essential for the removal of damaged mitochondria [23], and this action of HRW not only enhances self-repair capacity of the cell but also provides new insights into its potential clinical applications, particularly in the management of stress-related metabolic diseases.

These findings suggest that HRW could be an emerging strategy for treating chronic stress-related metabolic disorders. Given that these conditions often involve mitochondrial dysfunction, the application of HRW may help restore mitochondrial function, improve cellular metabolism, and ultimately enhance patients' quality of life. Future research should focus on elucidating the protective mechanisms of HRW in different pathological states and conducting clinical trials to validate its efficacy in managing stress-related diseases.

### 4. The effect of HRW on mitochondrial pathway apoptosis

Our findings demonstrate that HRW can significantly mitigate stress-induced apoptosis in liver cells by modulating the mitochondrial apoptotic pathway. The balance between pro-apoptotic and anti-apoptotic factors is crucial for cell survival under stress [24]. We observed an increase in Bcl-2 expression and a decrease in the release of cytochrome c (Cytc), indicating the key role of HRW in protecting against apoptosis. This aligns with prior research showing that hydrogen-rich solutions can inhibit apoptosis through similar mechanisms [25]. This finding not only validates the anti-apoptotic effects of HRW but also underscores its potential value as a clinical therapeutic strategy. Oxidative stress is recognized as a pathological mechanism underlying various diseases, including liver disease, cardiovascular conditions, and neurodegenerative disorders. Thus, HRW, as an intervention capable of enhancing anti-apoptotic signaling pathways, may be clinically applicable in reducing cell death caused by oxidative stress, ultimately improving patient outcomes [26].

Moreover, the antioxidant effects of HRW complement the current clinical practices involving antioxidants. This suggests that HRW could be used in conjunction with existing

treatment regimens to enhance therapeutic efficacy. For example, in the management of chronic liver disease, the use of HRW may help mitigate liver cell apoptosis induced by drugs or pathological conditions, thereby improving liver function and quality of life for patients [27, 28].

To further validate the role of HRW in apoptosis regulation, future research should explore its mechanisms of action, including how it modulates specific signaling pathways and intracellular environments. Moreover, when translating these findings into clinical practice, it is crucial to consider patient variability and the long-term safety of HRW to achieve optimal outcomes in the treatment of liver diseases.

## 5. The effect of HRW on lipid peroxidation and ferroptosis

Chronic stress often leads to increased lipid peroxidation due to excessive ROS production, making cells vulnerable to ferroptosis [29]. Our study revealed that HRW significantly reduces markers of lipid peroxidation, such as MDA and 4-HNE, suggesting its role in protecting against oxidative damage [30]. This finding is consistent with previous research indicating that hydrogen molecules can effectively neutralize ROS and reduce lipid peroxidation [31]. Moreover, the modulation of ferroptosis-related proteins, including GPX4 and SLC7A11, underscores HRW's ability to protect against ferroptosis, a regulated cell death pathway linked to oxidative stress [32]. Our study aligns with the growing body of research that highlights the role of Nrf2 in mediating cellular defense mechanisms against oxidative stress and ferroptosis [33]. The observed upregulation of Nrf2 and its target gene HO-1 in HRW-treated mice further supports the hypothesis that HRW activates endogenous antioxidant pathways to mitigate oxidative damage [34]. Clinically, these findings suggest that HRW may have potential applications in the treatment of metabolic diseases related to oxidative stress. Conditions associated with chronic stress, such as cardiovascular diseases, diabetes, and liver disorders, often involve oxidative damage and ferroptosis. Thus, the use of HRW may help alleviate these pathological processes, improving overall health outcomes for patients. Future research should further explore the role and mechanisms of HRW in various disease models to confirm its efficacy and safety for clinical applications.

Furthermore, while we have identified that HRW alleviates intestinal inflammation in mice with chronic inflammation through the Nf-κB pathway [35], it remains pertinent to explore whether, in the context of mice subjected to chronic stress, the amelioration of stress is achieved by modulating the Nf-κB pathway in the liver. This could provide a deeper understanding of how HRW confers its protective effects and its potential therapeutic applications.

## Conclusions

This study reveals that long-term hydrogen-rich water consumption provides significant hepatoprotection in mice under chronic stress. HRW normalizes liver enzyme levels, enhances antioxidant capacity, and reduces lipid peroxidation and ferroptosis. It improves mitochondrial biogenesis, function, and ATP production, and attenuates apoptosis by modulating related proteins. Behavioral tests show HRW alleviates stress-induced anxiety and enhances exploratory behavior. These findings suggest HRW is a promising non-invasive intervention for preventing and treating stress-related liver disorders by targeting oxidative stress and mitochondrial dysfunction.

## Supporting information

**S1 Table. Raw data of this article.**
(XLSX)

**S1 Raw images.**
(PDF)

## Author Contributions

**Conceptualization:** Qi He.

**Data curation:** Qi He.

**Formal analysis:** Xiang Lan.

**Funding acquisition:** Na Zhang.

**Investigation:** Xiang Lan, Mengyuan Ding, Na Zhang.

**Methodology:** Qi He.

**Project administration:** Qi He.

**Resources:** Xiang Lan, Na Zhang.

**Software:** Mengyuan Ding.

**Supervision:** Na Zhang.

**Validation:** Qi He, Mengyuan Ding.

**Visualization:** Qi He.

**Writing – original draft:** Qi He.

**Writing – review & editing:** Na Zhang.

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
