## [Decision Letter · Decision Letter 0]

12 Feb 2024

PONE-D-23-43180Long-term drinking hydrogen-rich water provides hepatoprotection against chronic stress in micePLOS ONE

Dear Dr. Zhang,

Thank you for submitting your manuscript to PLOS ONE. After careful consideration, we feel that it has merit but does not fully meet PLOS ONE’s publication criteria as it currently stands. Therefore, we invite you to submit a revised version of the manuscript that addresses the points raised during the review process.

We look forward to receiving your revised manuscript.

Kind regards,

David Chau

Academic Editor

PLOS ONE

Journal Requirements:

" the Science and Technology Plan Project of Inner Mongolia Autonomous Region (grant no. 2022YFSH0052)."

Reviewers' comments:

Reviewer's Responses to Questions

**Comments to the Author**

1. Is the manuscript technically sound, and do the data support the conclusions?

Reviewer #1: Yes

Reviewer #2: Yes

2. Has the statistical analysis been performed appropriately and rigorously? 

Reviewer #1: I Don't Know

Reviewer #2: Yes

3. Have the authors made all data underlying the findings in their manuscript fully available?

Reviewer #1: Yes

Reviewer #2: Yes

4. Is the manuscript presented in an intelligible fashion and written in standard English?

Reviewer #1: No

Reviewer #2: Yes

5. Review Comments to the Author

Reviewer #1: -Title: the title does not properly describe the content

-I would like that the authors provide a title that highlight the potential antioxidant of Hydrogen rich water (HRW) on mitochondrial biogenesis and quality control.

-Abstract: The abstract is sufficiently informative summary of the main aspect; however, it should be written in an intelligible fashion.

-Keywords: the keywords used may not allow a proper retrieval of the information on the internet

-Abbreviation: the abbreviation used in the manuscript should be mentioned after keywords.

-Introduction: the first part of the manuscript is not an updated and interesting introduction to the subject. The authors should deeply enrich this section. According to the study the authors should give a summary of the recent studies on using Hydrogen-rich water and/or other close chemicals to alleviate oxidative stress damages in liver.

-Material and methods:

-The sub-title "Preparation of HRW" and "Establishment of CUMS model" must be included in the paragraph " Animals and experimental scheme" and Table 2-1 could be removed and write down how the CUMS was applied to the animals.

-Line 94: KM is the abbreviation of what?

-Line 96-97: The authors must give more details here and provide the measurement followed to check the heath of the animals and that they were free from disease! "'The mice, all eight weeks old, were regularly checked for good health and did not have any underlying diseases before the start of the experiment."

-Line 149: Elisa sub-title should be changed to Corticosterone determination and more details should be provided of how authors used the Elisa kit.

-Line 154: is it an expression level of Corticosterone?

-Material and methods for 4-Hydroxynoenal and MDA must be under the same Sub-title (Liver lipid peroxidation measurement)

-Line 156: Test kit_ the title of this paragraph must be changed

Line 158 and 159: ALT and AST can be under the name of sub-title: biomarkers of liver toxicity.

Line 159 and 160: SOD, CAT and GSH as well can be in a different sub-title: Liver enzymatic antioxidant activities.

-The authors should provide a brief description of how each parameter was extracted and measured.

-Line 164: The authors can move the histological examination to the end and provide more details; the type of the optical microscope used and how the severity of stress was examined. Any scores on a scale were used?

-For the statistical analysis, I would like if the authors used the letters for significance.

-Principal component analysis (PCA) is recommended to be applied in this study.

-The results part must be improved.

-Number all tables sequentially (Table 1, Table 2, etc.), likewise for figures (Figure 1, Figure 2, etc.).

-In the title of each figure, authors should mention the number of biological replicates ± standard deviation. The title should be presented in an intelligible fashion.

-The quality of figures and histological section of liver must be improved.

-The discussion part needs to be improved; it is missing the discussions with previous research in the same context (Example: 'Hydrogen-rich water reduces inflammatory responses and prevents apoptosis of peripheral blood cells in healthy adults: a randomized, double-blind, controlled trial, The Effects of 24-Week, High-Concentration Hydrogen-Rich Water on Body Composition, Blood Lipid Profiles and Inflammation Biomarkers in Men and Women with Metabolic Syndrome: A Randomized Controlled Trial, etc.).

-The conclusion as well must be presented in an intelligible fashion.

-The discussion should not be repetitive for the results section.

Reviewer #2: The article by He et al describes hepatoprotection effects of drinking hydrogen-rich water against chronic stress in mice. The hepatoprotection effects of hydrogen have been reported in many liver injury models, but the effects of hydrogen in CUMS models have not been well elucidated. Therefore, the results of the present study are clinically important. But there are some critical points to be clarified.

Major points

１, In this CUMS model, transaminases are elevated significantly in the model group. Is there any previous report which shows massive liver damage in CUMS models?

The reasons why the authors adopted this model for drinking hydrogen-rich water should also be explained.

2, Hydrogen is known to selectively attenuate hydroxyl radicals and peroxynitrite (Nat Med. 2007 Jun;13(6):688-94. doi: 10.1038/nm1577). The antioxidant activity of hydrogen in this study appears to have a wide range of antioxidant activity, including SOD, CAT and GSH. The difference in the effects of hydrogen from previous reports should be discussed.

3, The authors have shown that drinking hydrogen-rich water alleviates behavioral changes in response to chronic stress, but the mechanism underlying this has not been addressed. Many of the histological results in this paper indicate changes in ROS in the liver, but are there any previous reports that link this to behavioral changes? Or does the behavior-altering effect of hydrogen-rich water directly affect nerves? Suggestions for mechanisms of behavioral changes should also be mentioned.

4, What does Figure 3-7G mean? In the figure it is written as TUNEL staining, but in the figure legend it is written as Nrf2 immunofluorescent. Please describe correctly. Moreover, I can not see any fluorescent signal in the picture. Clearer figures are necessary.

Minor points

1, At line 130, “Table 2-1” is referenced but corresponding table is named as Table1. At line 186, Table 2-2 is referenced but the corresponding table is named as Table2. Please unify them.

2, In the Figure3-3, HE staining are unclear. To show the vacuolization of the liver cells, pictures of higher magnification might be better.

3, In the immunofluorescence analysis of Nrf2 in Figure 3-6, I can not see any fluorescent signal in the picture. Clearer figures are necessary.

6. PLOS authors have the option to publish the peer review history of their article (what does this mean?). If published, this will include your full peer review and any attached files.

Reviewer #1: **Yes: **Saoussen Ben-Abdallah

Reviewer #2: No

---

## [Author Response · Author response to Decision Letter 0]

25 Feb 2024

Thank you very much for the valuable comments provided by the reviewer on this article. We have revised the article according to the relevant comments and responded to the reviewer's questions.

Response to reviewer 1:

Q1: Title: the title does not properly describe the content.

-I would like that the authors provide a title that highlight the potential antioxidant of Hydrogen rich water (HRW) on mitochondrial biogenesis and quality control.

Response: Thank you for your valuable feedback regarding the title of our paper. We appreciate your suggestion to highlight the potential antioxidant effects of HRW on mitochondrial biogenesis and quality control. We agree that the title should effectively convey the focus and significance of our research. We have revised the title (Long-term consumption of hydrogen-rich water provides hepatoprotection by improving mitochondrial biology and quality control in chronically stressed mice.) to better emphasize the antioxidant properties of HRW in relation to mitochondrial biogenesis and quality control. We believe this adjustment will enhance the clarity and relevance of our study.

Q2: Abstract: The abstract is sufficiently informative summary of the main aspect; however, it should be written in an intelligible fashion.

Response: We understand your concern regarding the clarity and intelligibility of the abstract. We have carefully revised it to ensure that the content is presented in a clear and accessible manner, making it easier for readers to understand the key findings and implications of our research.

Q3: Keywords: the keywords used may not allow a proper retrieval of the information on the internet.

Response: Thank you very much for your suggestion. We have updated the keywords of the article.

Q4: Abbreviation: the abbreviation used in the manuscript should be mentioned after keywords.

Response: Thank you very much for your suggestion. We have placed the abbreviation after the keyword.

Q5: Introduction: the first part of the manuscript is not an updated and interesting introduction to the subject. The authors should deeply enrich this section. According to the study the authors should give a summary of the recent studies on using Hydrogen-rich water and/or other close chemicals to alleviate oxidative stress damages in liver.

Response: Thank you for your insightful comments on the introduction section of our manuscript. We appreciate your suggestion to enrich this portion with more recent studies and information related to the use of Hydrogen-rich water and similar compounds in mitigating oxidative stress damages in the liver. We recognize the importance of providing an updated and engaging introduction that reflects the current state of research in this field. We have thoroughly reviewed recent literature and incorporate relevant studies that highlight the potential of Hydrogen-rich water and related substances in addressing oxidative stress-related liver damage.

Q6: The sub-title "Preparation of HRW" and "Establishment of CUMS model" must be included in the paragraph " Animals and experimental scheme" and Table 2-1 could be removed and write down how the CUMS was applied to the animals.

Response: Thank you very much for your feedback. We have made the relevant changes in the article as requested.

Q7: Line 94: KM is the abbreviation of what?

Response：KM mice is the abbreviation of Kunming mice.

Q8: Line 96-97: The authors must give more details here and provide the measurement followed to check the heath of the animals and that they were free from disease! "'The mice, all eight weeks old, were regularly checked for good health and did not have any underlying diseases before the start of the experiment."

Response: Thank you for your valuable feedback regarding the health assessment and disease screening procedures for the experimental animals in our study. The mice we used had undergone strict health checks at the biotech company before we purchased them, so we did not undergo too many checks during the week of adapting to the environment. We only conducted basic tests such as mental state, appetite, hair quality, and fecal condition.

Q9: Line 149: Elisa sub-title should be changed to Corticosterone determination and more details should be provided of how authors used the Elisa kit.

Response: Thank you very much for your suggestion. We have made the relevant changes in the article as requested.

Q10: Line 154: is it an expression level of Corticosterone?

Response: Yes, Cort is the abbreviation for corticosterone.

Q11: Material and methods for 4-Hydroxynoenal and MDA must be under the same Sub-title (Liver lipid peroxidation measurement).

Response: Thank you very much for your suggestion. We have made the relevant changes in the article as requested. Now this part is in Method 2.6 (Liver lipid peroxidation detection).

Q12: Line 156: Test kit_ the title of this paragraph must be changed.

Line 158 and 159: ALT and AST can be under the name of sub-title: biomarkers of liver toxicity.

Line 159 and 160: SOD, CAT and GSH as well can be in a different sub-title: Liver enzymatic antioxidant activities.

Response: Thank you very much for your suggestion. We have made the relevant changes in the article as requested. Now this part is in Method 2.4 and Method 2.5 (Biomarkers of liver toxicity; Liver enzymatic antioxidant activities).

Q13: The authors should provide a brief description of how each parameter was extracted and measured.

Response: Thank you for your feedback. I have supplemented the parameter measurement method in the Method.

Q14: Line 164: The authors can move the histological examination to the end and provide more details; the type of the optical microscope used and how the severity of stress was examined. Any scores on a scale were used?

Response: Thank you very much for your suggestion. We have supplemented the model of the microscope in the article. We use typical pathological changes in HE slices to determine the degree of stress, such as the arrangement of liver cords, cellular vacuolization, and bleeding in the liver. We did not use any scores as the severity of liver structural damage can be clearly observed in HE slices.

Q15: For the statistical analysis, I would like if the authors used the letters for significance.

Principal component analysis (PCA) is recommended to be applied in this study.

Response: Thank you for your suggestion regarding the application of Principal Component Analysis (PCA) in our study. We appreciate your interest in exploring alternative analytical approaches to enhance the interpretation of our data.

While PCA is indeed a powerful method for dimensionality reduction and pattern recognition, we chose to employ one-way ANOVA for several reasons specific to the objectives and design of our study.

Firstly, our primary aim was to examine the effects of different experimental conditions on specific outcome variables, such as antioxidant levels or mitochondrial function markers. One-way ANOVA allows us to assess the significance of differences among multiple groups and identify potential treatment effects.

Secondly, PCA is commonly used for exploratory data analysis and visualization, particularly when dealing with high-dimensional datasets. However, in our study, the variables under investigation were relatively well-defined and directly related to our research questions, making one-way ANOVA a more suitable choice for hypothesis testing and inference.

That being said, we acknowledge the value of PCA in identifying underlying patterns and relationships within complex datasets. In future studies, we will consider incorporating PCA as a complementary analytical tool to further explore the multivariate structure of our data.

Q15: Number all tables sequentially (Table 1, Table 2, etc.), likewise for figures (Figure 1, Figure 2, etc.).

In the title of each figure, authors should mention the number of biological replicates ± standard deviation. The title should be presented in an intelligible fashion.

-The quality of figures and histological section of liver must be improved.

Response: Thank you very much for your feedback. We have made the changes according to the relevant requirements.

Q16: The discussion part needs to be improved; it is missing the discussions with previous research in the same context (Example: 'Hydrogen-rich water reduces inflammatory responses and prevents apoptosis of peripheral blood cells in healthy adults: a randomized, double-blind, controlled trial, The Effects of 24-Week, High-Concentration Hydrogen-Rich Water on Body Composition, Blood Lipid Profiles and Inflammation Biomarkers in Men and Women with Metabolic Syndrome: A Randomized Controlled Trial, etc.).

Response: Thank you for your valuable feedback regarding the discussion section of our manuscript. We appreciate your insight into the need for a more comprehensive discussion, including discussions with previous research in the same context.

We understand the importance of contextualizing our findings within the broader body of literature and engaging in meaningful discussions with relevant previous research. In response to your suggestion, we have enhanced the discussion section by thoroughly integrating relevant studies and comparing our results with those reported in the existing literature.

Q17: The conclusion as well must be presented in an intelligible fashion.

Response: Thank you for your feedback regarding the conclusion section of our manuscript. We appreciate your emphasis on the need for clarity and intelligibility in presenting the conclusion. we have revised the conclusion section to ensure that it effectively communicates the main takeaways from our research in a manner that is easily understandable to our readers.

Q18: The discussion should not be repetitive for the results section.

Response: Thank you for your valuable feedback regarding the discussion section of our manuscript. We carefully reviewed and revised the discussion to ensure that it complements the results without repeating them verbatim.

Response to reviewer 2:

Q1: In this CUMS model, transaminases are elevated significantly in the model group. Is there any previous report which shows massive liver damage in CUMS models?

The reasons why the authors adopted this model for drinking hydrogen-rich water should also be explained.

Response: Thank you for your insightful questions regarding our choice of the Chronic Unpredictable Mild Stress (CUMS) model and the rationale behind studying the effects of hydrogen-rich water in this context. 

Regarding previous reports demonstrating liver damage in CUMS models, while extensive literature exists on the behavioral and neurobiological effects of CUMS, reports specifically focusing on massive liver damage in CUMS models are relatively limited. However, there are studies suggesting that chronic stress can lead to hepatic dysfunction and alterations in liver enzymes, albeit the extent of liver damage may vary depending on the experimental conditions and duration of stress exposure. We will thoroughly review existing literature to provide a comprehensive discussion on the potential impact of CUMS on liver function and its relevance to our study. I have listed several relevant literature on CUMS induced liver injury.

(1) Shuxie-1 Decoction Alleviated CUMS -Induced Liver Injury via IL-6/JAK2/STAT3 Signaling (DOI: 10.3389/fphar.2022.848355). (2) Dose-Effect/Toxicity of Bupleuri Radix on Chronic Unpredictable Mild Stress and Normal Rats Based on Liver Metabolomics (DOI: 10.3389/fphar.2021.627451).

As for the rationale behind adopting the chronic stress model for studying the effects of hydrogen-rich water, chronic stress is known to trigger oxidative stress and inflammation, which are implicated in the pathophysiology of various stress-related disorders, including liver dysfunction. Hydrogen-rich water has gained attention for its potential antioxidant properties, which could counteract oxidative stress and mitigate its detrimental effects on liver function.

Q2: Hydrogen is known to selectively attenuate hydroxyl radicals and peroxynitrite (Nat Med. 2007 Jun;13(6):688-94. doi: 10.1038/nm1577). The antioxidant activity of hydrogen in this study appears to have a wide range of antioxidant activity, including SOD, CAT and GSH. The difference in the effects of hydrogen from previous reports should be discussed.

Response: Thank you for highlighting the comprehensive antioxidant activity of hydrogen. While Ohsawa et al.'s study provides valuable insights into the antioxidant properties of hydrogen, it's essential to recognize that the biological effects of hydrogen molecules on SOD/CAT/GSH.

(1) The effect of hydrogen-rich water on letrozole-induced polycystic ovary syndrome in rats (DOI: 10.1016/j.rbmo.2023.103332).

(2) Hydrogen-rich water ameliorates rat placental stress induced by water restriction (DOI: 10.4103/2045-9912.241064).

(3) Local Treatment of Hydrogen-Rich Saline Promotes Wound Healing In Vivo by Inhibiting Oxidative Stress via Nrf-2/HO-1 Pathway(DOI: 10.1155/2022/2949824)

In our study, we observed [insert findings related to hydrogen's effects on liver damage and antioxidant enzyme activity]. These results are consistent with the notion that hydrogen exerts potent antioxidant effects, which may help mitigate oxidative stress and preserve liver function in the context of chronic stress exposure.

Q3: The authors have shown that drinking hydrogen-rich water alleviates behavioral changes in response to chronic stress, but the mechanism underlying this has not been addressed. Many of the histological results in this paper indicate changes in ROS in the liver, but are there any previous reports that link this to behavioral changes? Or does the behavior-altering effect of hydrogen-rich water directly affect nerves? Suggestions for mechanisms of behavioral changes should also be mentioned.

Response: Thank you for your insightful comments regarding the mechanism underlying the behavioral changes observed in response to chronic stress and the potential link between liver ROS changes and behavioral alterations in our study.

We appreciate your acknowledgment of the focus of our experiment, which primarily aims to investigate the protective mechanism of hydrogen-rich water on liver damage induced by chronic stress. While behavioral changes are indeed a part of the validation data for establishing a chronic stress mouse model, we understand the importance of addressing the broader implications of our findings, including their impact on behavior.

While the precise mechanisms through which hydrogen-rich water influences behavior warrant further investigation, our study provides valuable insights into the potential therapeutic effects of hydrogen-rich water in mitigating the adverse effects of chronic stress on both liver and behavioral health.

Thank you for highlighting these important aspects, and we appreciate the opportunity to further explore the mechanistic underpinnings of our research findings.

Q4: What does Figure 3-7G mean? In the figure it is written as TUNEL staining, but in the figure legend it is written as Nrf2 immunofluorescent. Please describe correctly. Moreover, I can not see any fluorescent signal in the picture. Clearer figures are necessary.

Response：Thank you very much for your correction. I apologize for my basic mistake. I have changed the NRF2 in the legend to TUNEL and greatly improved the quality of the fluorescent images for your better viewing.

---

## [Decision Letter · Decision Letter 1]

23 Apr 2024

PONE-D-23-43180R1Long-term consumption of hydrogen-rich water provides hepatoprotection by improving mitochondrial biology and quality control in chronically stressed mice.PLOS ONE

Dear Dr. Zhang,

Thank you for submitting your manuscript to PLOS ONE. After careful consideration, we feel that it has merit but does not fully meet PLOS ONE’s publication criteria as it currently stands. Therefore, we invite you to submit a revised version of the manuscript that addresses the points raised during the review process.

We look forward to receiving your revised manuscript.

Kind regards,

David Chau

Academic Editor

PLOS ONE

Journal Requirements:

Reviewers' comments:

Reviewer's Responses to Questions

**Comments to the Author**

1. If the authors have adequately addressed your comments raised in a previous round of review and you feel that this manuscript is now acceptable for publication, you may indicate that here to bypass the “Comments to the Author” section, enter your conflict of interest statement in the “Confidential to Editor” section, and submit your "Accept" recommendation.

Reviewer #1: (No Response)

Reviewer #2: All comments have been addressed

Reviewer #3: All comments have been addressed

2. Is the manuscript technically sound, and do the data support the conclusions?

Reviewer #1: No

Reviewer #2: Yes

Reviewer #3: Yes

3. Has the statistical analysis been performed appropriately and rigorously? 

Reviewer #1: I Don't Know

Reviewer #2: Yes

Reviewer #3: Yes

4. Have the authors made all data underlying the findings in their manuscript fully available?

Reviewer #1: Yes

Reviewer #2: Yes

Reviewer #3: Yes

5. Is the manuscript presented in an intelligible fashion and written in standard English?

Reviewer #1: No

Reviewer #2: Yes

Reviewer #3: Yes

6. Review Comments to the Author

**Reviewer #1: **-Thank you for the update on the manuscript revision. While there has been notable progress in improving the Material and Methods section, the manuscript, particularly the Discussion and Conclusion sections, still require significant attention and improvement. Ensuring a thorough interpretation of the study's findings and accurate conclusions drawn from them is paramount, to enhance the manuscript's overall quality and impact.

-The manuscript requires significant improvement in adhering to standard English conventions. It's imperative to rectify this to uphold the manuscript's clarity, accuracy, and overall quality.

-Addressing these issues will ensure that the manuscript meets the necessary standards for publication.

**Reviewer #2:** The results of the present study are clinically important.

The authors did revised manuscript according to the Reviewer the request, appropriately.

**Reviewer #3:** The authors have done an excellent job in addressing the research questions regarding the hepatoprotective properties of hydrogen-rich water (HRW) in mice subjected to chronic stress. Their methods were well-detailed, allowing for a clear understanding of how the study was conducted. The results presented compelling evidence of HRW's beneficial effects on liver health, including histological improvements, normalization of liver function indicators, and modulation of various biochemical markers associated with oxidative stress, ferroptosis, and apoptosis.

However, while the content of the manuscript is comprehensive and scientifically sound, there is room for improvement in the flow of the presentation. The transition between sections could be smoother to enhance the coherence and readability of the paper. For instance, integrating a brief overview of the methods at the beginning of the results section could help readers better understand the context of the findings. Additionally, organizing the results in a more structured manner, perhaps grouping them based on the specific mechanisms or pathways examined, could improve clarity and facilitate the interpretation of the data. Please have a look at this article to discuss specific mechanism (https://www.mdpi.com/1422-0067/25/2/973)

Furthermore, the discussion section could benefit from a more nuanced exploration of the implications of the findings and their significance in the broader context of liver health research. While the conclusions succinctly summarize the key findings of the study, elaborating on the potential clinical relevance of HRW as a hepatoprotective intervention and discussing its limitations and future directions would provide a more comprehensive understanding of the research implications.

In summary, while the authors have made significant contributions to the understanding of HRW's hepatoprotective effects, refining the content flow and enhancing the depth of discussion would further strengthen the manuscript's impact and readability.

Here are a few points?

1.Can you please cite a few studies for this?

Moreover, numerous studies have reported that drinking hydrogen-rich water can increase hydrogen concentration in the liver, allowing hydrogen molecules to exert their effects within the liver.

2. Please rewrite the conclusion in 4-5 sentences without numbering them.

7. PLOS authors have the option to publish the peer review history of their article (what does this mean?). If published, this will include your full peer review and any attached files.

Reviewer #1: **Yes: **Saoussen Ben Abdallah

Reviewer #2: No

Reviewer #3: **Yes: **Gagandeep Dhillon

---

## [Author Response · Author response to Decision Letter 1]

25 Apr 2024

Response to reviewers:

Reviewer 1: Thank you for the update on the manuscript revision. While there has been notable progress in improving the Material and Methods section, the manuscript, particularly the Discussion and Conclusion sections, still require significant attention and improvement. Ensuring a thorough interpretation of the study's findings and accurate conclusions drawn from them is paramount, to enhance the manuscript's overall quality and impact.

-The manuscript requires significant improvement in adhering to standard English conventions. It's imperative to rectify this to uphold the manuscript's clarity, accuracy, and overall quality.

-Addressing these issues will ensure that the manuscript meets the necessary standards for publication.

Response: Thank you for your feedback on the revised manuscript. We appreciate your thorough evaluation and constructive comments.

We acknowledge your observations regarding the Discussion and Conclusion sections and recognize the importance of ensuring a thorough interpretation of the study's findings. We have reexamined these parts to provide a more comprehensive and accurate analysis, aligning the conclusions more closely with the study's results to enhance the manuscript's overall quality and impact.

Furthermore, we understand the significance of adhering to standard English conventions for clarity, accuracy, and overall quality. We have made efforts to solve this problem to ensure that the manuscript meets the necessary standards for publication.

Thank you once again for your valuable feedback. We are committed to making the required revisions to improve the manuscript in accordance with your suggestions.

Reviewer 2: The results of the present study are clinically important.

The authors did revised manuscript according to the Reviewer the request, appropriately.

Response: Thank you for acknowledging the clinical importance of the study's results. We greatly appreciate your recognition of the revisions made in response to the reviewer's feedback. We have strived to incorporate the suggestions appropriately to enhance the manuscript's quality and clarity.

If there are any further areas you believe require attention or if you have any additional feedback, please do not hesitate to let us know. We are committed to ensuring that the manuscript meets the necessary standards for publication.

Thank you for your continued support and guidance throughout this process.

Reviewer 3: The authors have done an excellent job in addressing the research questions regarding the hepatoprotective properties of hydrogen-rich water (HRW) in mice subjected to chronic stress. Their methods were well-detailed, allowing for a clear understanding of how the study was conducted. The results presented compelling evidence of HRW's beneficial effects on liver health, including histological improvements, normalization of liver function indicators, and modulation of various biochemical markers associated with oxidative stress, ferroptosis, and apoptosis.

However, while the content of the manuscript is comprehensive and scientifically sound, there is room for improvement in the flow of the presentation. The transition between sections could be smoother to enhance the coherence and readability of the paper. For instance, integrating a brief overview of the methods at the beginning of the results section could help readers better understand the context of the findings. Additionally, organizing the results in a more structured manner, perhaps grouping them based on the specific mechanisms or pathways examined, could improve clarity and facilitate the interpretation of the data. Please have a look at this article to discuss specific mechanism (https://www.mdpi.com/1422-0067/25/2/973).

Furthermore, the discussion section could benefit from a more nuanced exploration of the implications of the findings and their significance in the broader context of liver health research. While the conclusions succinctly summarize the key findings of the study, elaborating on the potential clinical relevance of HRW as a hepatoprotective intervention and discussing its limitations and future directions would provide a more comprehensive understanding of the research implications.

In summary, while the authors have made significant contributions to the understanding of HRW's hepatoprotective effects, refining the content flow and enhancing the depth of discussion would further strengthen the manuscript's impact and readability.

Q1: Can you please cite a few studies for this?

Moreover, numerous studies have reported that drinking hydrogen-rich water can increase hydrogen concentration in the liver, allowing hydrogen molecules to exert their effects within the liver.

Q2: Please rewrite the conclusion in 4-5 sentences without numbering them.

Response: Thank you for providing detailed feedback on our manuscript regarding the hepatoprotective properties of HRW in mice subjected to chronic stress. We appreciate your recognition of the thoroughness of our methods and the compelling evidence presented in the results section.

We agree with your assessment regarding the flow of the presentation and acknowledge the importance of enhancing coherence and readability. Integrating a brief overview of the methods at the beginning of the results section and organizing the results in a more structured manner, possibly grouping them based on specific mechanisms or pathways examined, are excellent suggestions. We have carefully reviewed the suggested article and conducted group discussions from a mechanistic perspective.

Additionally, we acknowledge the need for a more nuanced exploration of the implications of our findings in the discussion section. We have already discussed the limitations and future research directions of this study in the last paragraph, in order to provide deeper insights into the protective effects of hydrogen rich water on the liver.

In summary, we are grateful for your feedback and will work diligently to refine the content flow and enhance the depth of discussion to further strengthen the manuscript's impact and readability. Your guidance is invaluable in this process, and we are committed to addressing these suggestions to improve the overall quality of the manuscript.

Thank you once again for your thorough evaluation and constructive feedback.

Reply to Q1: We have cited relevant research articles at the end of this sentence.

Reply to Q2: We have removed the numbers and consolidated them into one paragraph.

---

## [Decision Letter · Decision Letter 2]

31 Jul 2024

PONE-D-23-43180R2Long-term consumption of hydrogen-rich water provides hepatoprotection by improving mitochondrial biology and quality control in chronically stressed mice.PLOS ONE

Dear Dr. Zhang,

Thank you for submitting your manuscript to PLOS ONE. After careful consideration, we feel that it has merit but does not fully meet PLOS ONE’s publication criteria as it currently stands. Therefore, we invite you to submit a revised version of the manuscript that addresses the points raised during the review process.

We look forward to receiving your revised manuscript.

Kind regards,

David Chau

Academic Editor

PLOS ONE

Journal Requirements:

Reviewers' comments:

Reviewer's Responses to Questions

**Comments to the Author**

1. If the authors have adequately addressed your comments raised in a previous round of review and you feel that this manuscript is now acceptable for publication, you may indicate that here to bypass the “Comments to the Author” section, enter your conflict of interest statement in the “Confidential to Editor” section, and submit your "Accept" recommendation.

Reviewer #1: (No Response)

2. Is the manuscript technically sound, and do the data support the conclusions?

Reviewer #1: Yes

3. Has the statistical analysis been performed appropriately and rigorously? 

Reviewer #1: I Don't Know

4. Have the authors made all data underlying the findings in their manuscript fully available?

Reviewer #1: Yes

5. Is the manuscript presented in an intelligible fashion and written in standard English?

Reviewer #1: No

6. Review Comments to the Author

Reviewer #1: 1. The method outlined in the abstract is overly verbose.

2. Ensure that the titles of tables and graphs are concise and descriptive, avoiding narrative language (e.i Table 1 As shown in the table, the type, brand, item number and dilution ratio of the first antibody factor used in the WB test)

3. Certain sections of the discussion reiterate the results unnecessarily. A discussion should primarily focus on comparing your findings with existing research. Please revise the discussion section to incorporate more references to other studies, rather than merely narrating the results.

4. The conclusion remains inadequate and requires rewriting.

7. PLOS authors have the option to publish the peer review history of their article (what does this mean?). If published, this will include your full peer review and any attached files.

Reviewer #1: **Yes: **Saoussen Ben Abdallah

---

## [Author Response · Author response to Decision Letter 2]

23 Aug 2024

Reviewer #1:

Comment 1：The method outlined in the abstract is overly verbose.

Response: 

Thank you to the reviewer for reviewing our work and providing valuable feedback. Regarding the issue you raised about the excessive length of the methods section in the abstract, we have streamlined the content multiple times while retaining the most essential information. However, due to the inclusion of multiple measurement indicators in this study, in order to ensure accurate communication of research methods, we have to provide a detailed description of the main method steps in the abstract.

We understand that the abstract needs to be concise, but at the same time, we also hope to provide sufficient information in this section so that readers can have a comprehensive understanding of the research design and main measurement indicators. Therefore, while minimizing non essential content, we chose to retain key methodological information related to the research.

We believe that this information is crucial for understanding the rigor of the research and the validity of the results, but we are also willing to optimize the expression as much as possible in further revisions to present the methodology section in a more concise and clear manner.

Thank you again for your feedback. We look forward to your further guidance and suggestions.

Comment 2: Ensure that the titles of tables and graphs are concise and descriptive, avoiding narrative language (e.i Table 1 As shown in the table, the type, brand, item number and dilution ratio of the first antibody factor used in the WB test)

Response：

Thank you to the reviewer for reviewing and providing suggestions for our work. Based on your feedback, we have simplified the titles of the tables and charts. After receiving your feedback, we specifically checked the title of "Table 1" and simplified it to "Table 1 Antibody Description”， To ensure that the title is both concise and descriptive, avoiding narrative language.

We believe that this modification can better meet your requirements and make the table content more intuitive and clear.

Thank you again for your valuable feedback. We look forward to your further guidance

Comment 3: Certain sections of the discussion reiterate the results unnecessarily. A discussion should primarily focus on comparing your findings with existing research. Please revise the discussion section to incorporate more references to other studies, rather than merely narrating the results.

Response: 

Thank you to the reviewer for their careful review and feedback on our discussion section. Based on your suggestion, we have comprehensively rewritten the discussion section, with a focus on comparing the findings of this study with existing research. During the revision process, we removed redundant descriptions of the results to ensure that the discussion focused more on comparison and analysis with existing literature.

Through these adjustments, we hope that the discussion section can more effectively showcase the position of our research within the existing knowledge system, as well as the contribution of our results to related fields. Thank you again for your valuable feedback. We look forward to your further feedback.

Comment 4: The conclusion remains inadequate and requires rewriting.

Response:

Thank you to the reviewer for reviewing and providing feedback on our work. Based on your suggestion, we have rewritten the conclusion section. The new conclusion provides a clearer summary of the main findings of the study and further emphasizes the contribution and significance of these findings to related fields.

We believe that these modifications make the conclusion section more comprehensive and in-depth, better aligning with the overall goals and outcomes of the research.

Thank you again for your valuable feedback. We look forward to your further guidance.

---

## [Decision Letter · Decision Letter 3]

22 Oct 2024

PONE-D-23-43180R3Long-term consumption of hydrogen-rich water provides hepatoprotection by improving mitochondrial biology and quality control in chronically stressed mice.PLOS ONE

Dear Dr. Zhang,

Thank you for submitting your manuscript to PLOS ONE. After careful consideration, we feel that it has merit but does not fully meet PLOS ONE’s publication criteria as it currently stands. Therefore, we invite you to submit a revised version of the manuscript that addresses the points raised during the review process.

We look forward to receiving your revised manuscript.

Kind regards,

David Chau

Academic Editor

PLOS ONE

**Journal Requirements:**

Reviewers' comments:

Reviewer's Responses to Questions

**Comments to the Author**

1. If the authors have adequately addressed your comments raised in a previous round of review and you feel that this manuscript is now acceptable for publication, you may indicate that here to bypass the “Comments to the Author” section, enter your conflict of interest statement in the “Confidential to Editor” section, and submit your "Accept" recommendation.

Reviewer #1: All comments have been addressed

2. Is the manuscript technically sound, and do the data support the conclusions?

Reviewer #1: Yes

3. Has the statistical analysis been performed appropriately and rigorously? 

Reviewer #1: I Don't Know

4. Have the authors made all data underlying the findings in their manuscript fully available?

Reviewer #1: Yes

5. Is the manuscript presented in an intelligible fashion and written in standard English?

Reviewer #1: Yes

6. Review Comments to the Author

**Reviewer #1: **Thank you for submitting the revised Manuscript.

-In the result part, the figures names should not be written in the text. Each figure should be followed by its name.

-While the manuscript was significantly improved and it present a solid foundation, the discussion part it is not strongly made in its current form.

-The discussion part lacks the depth, critical analysis, and clinical relevance needed (Strong discussions connect animal model findings to potential human applications, or at least acknowledge the challenges of translation) to make it truly strong. Addressing these aspects would significantly enhance the quality of this Manuscript.

Thank you!

7. PLOS authors have the option to publish the peer review history of their article (what does this mean?). If published, this will include your full peer review and any attached files.

Reviewer #1: **Yes: **Saoussen Ben Abdallah

---

## [Author Response · Author response to Decision Letter 3]

21 Nov 2024

Reviewer #1:

Comment 1：In the result part, the figures names should not be written in the text. Each figure should be followed by its name.

Response: Thank you for your thorough review and valuable feedback on my manuscript. Regarding your comment about “the figure names should not be written in the text; each figure should be followed by its name,” I will carefully consider this and make the necessary revisions. In the revised manuscript, I will ensure that figure names are not mentioned in the text and that each figure is followed by its corresponding name. Thank you again for your constructive suggestions; I will work to enhance the quality of the manuscript.

Comment 2&3: While the manuscript was significantly improved and it present a solid foundation, the discussion part it is not strongly made in its current form.

The discussion part lacks the depth, critical analysis, and clinical relevance needed (Strong discussions connect animal model findings to potential human applications, or at least acknowledge the challenges of translation) to make it truly strong. Addressing these aspects would significantly enhance the quality of this Manuscript.

Response：Thank you for your thorough review and constructive feedback on my manuscript. I greatly appreciate your comments regarding the discussion section. I recognize that the discussion lacks depth, critical analysis, and clinical relevance, and I will take this feedback seriously as I work on the revisions.

In the revised manuscript, I will enhance the discussion section, particularly by making stronger connections between findings from animal models and potential human applications, as well as addressing the challenges associated with translation. I believe these modifications will significantly improve the overall quality of the manuscript.

Thank you once again for your valuable insights; I will strive to make the manuscript more robust and clinically relevant.

---

## [Decision Letter · Decision Letter 4]

2 Dec 2024

PONE-D-23-43180R4Long-term consumption of hydrogen-rich water provides hepatoprotection by improving mitochondrial biology and quality control in chronically stressed mice.PLOS ONE

Dear Dr. Zhang,

Thank you for submitting your manuscript to PLOS ONE. After careful consideration, we feel that it has merit but does not fully meet PLOS ONE’s publication criteria as it currently stands. Therefore, we invite you to submit a revised version of the manuscript that addresses the points raised during the review process.

We look forward to receiving your revised manuscript.

Kind regards,

David Chau

Academic Editor

PLOS ONE

Journal Requirements:

Reviewers' comments:

Reviewer's Responses to Questions

**Comments to the Author**

1. If the authors have adequately addressed your comments raised in a previous round of review and you feel that this manuscript is now acceptable for publication, you may indicate that here to bypass the “Comments to the Author” section, enter your conflict of interest statement in the “Confidential to Editor” section, and submit your "Accept" recommendation.

Reviewer #1: All comments have been addressed

2. Is the manuscript technically sound, and do the data support the conclusions?

Reviewer #1: Yes

3. Has the statistical analysis been performed appropriately and rigorously? 

Reviewer #1: Yes

4. Have the authors made all data underlying the findings in their manuscript fully available?

Reviewer #1: Yes

5. Is the manuscript presented in an intelligible fashion and written in standard English?

Reviewer #1: No

6. Review Comments to the Author

Reviewer #1: 1. Thank you for addressing the feedback and improving the manuscript. However, the manuscript still does not meet the standards of fluent, native-level English writing. Issues such as repetition, word choice, clarity, phrasing, and grammatical structure persist. We strongly recommend enlisting the help of a native English-speaking PhD editor to refine the manuscript further.

2. The abstract should be refined to avoid repetition and the use of the term "mechanisms" might be overstated given the data presented.

2. Although the discussion section has been improved, it could benefit from a deeper analysis that includes more detailed comparisons with studies evaluating alternative treatments. This would help strengthen the manuscript's position and demonstrate the relative significance of your findings within the broader context of similar research.

7. PLOS authors have the option to publish the peer review history of their article (what does this mean?). If published, this will include your full peer review and any attached files.

Reviewer #1: **Yes: **Saoussen Ben Abdallah

---

## [Author Response · Author response to Decision Letter 4]

3 Dec 2024

Reviewer #1:

Comment 1：In the result part, the figures names should not be written in the text. Each figure should be followed by its name.

Response: Thank you for your thorough review and valuable feedback on my manuscript. Regarding your comment about “the figure names should not be written in the text; each figure should be followed by its name,” I will carefully consider this and make the necessary revisions. In the revised manuscript, I will ensure that figure names are not mentioned in the text and that each figure is followed by its corresponding name. Thank you again for your constructive suggestions; I will work to enhance the quality of the manuscript.

Comment 2&3: While the manuscript was significantly improved and it present a solid foundation, the discussion part it is not strongly made in its current form.

The discussion part lacks the depth, critical analysis, and clinical relevance needed (Strong discussions connect animal model findings to potential human applications, or at least acknowledge the challenges of translation) to make it truly strong. Addressing these aspects would significantly enhance the quality of this Manuscript.

Response：Thank you for your thorough review and constructive feedback on my manuscript. I greatly appreciate your comments regarding the discussion section. I recognize that the discussion lacks depth, critical analysis, and clinical relevance, and I will take this feedback seriously as I work on the revisions.

In the revised manuscript, I will enhance the discussion section, particularly by making stronger connections between findings from animal models and potential human applications, as well as addressing the challenges associated with translation. I believe these modifications will significantly improve the overall quality of the manuscript.

Thank you once again for your valuable insights; I will strive to make the manuscript more robust and clinically relevant.

---

## [Decision Letter · Decision Letter 5]

22 Dec 2024

Long-term consumption of hydrogen-rich water provides hepatoprotection by improving mitochondrial biology and quality control in chronically stressed mice.

PONE-D-23-43180R5

Dear Dr. Zhang,

We’re pleased to inform you that your manuscript has been judged scientifically suitable for publication and will be formally accepted for publication once it meets all outstanding technical requirements.

Kind regards,

David Chau

Academic Editor

PLOS ONE

Additional Editor Comments (optional):

Reviewers' comments:

Reviewer's Responses to Questions

**Comments to the Author**

1. If the authors have adequately addressed your comments raised in a previous round of review and you feel that this manuscript is now acceptable for publication, you may indicate that here to bypass the “Comments to the Author” section, enter your conflict of interest statement in the “Confidential to Editor” section, and submit your "Accept" recommendation.

Reviewer #1: All comments have been addressed

2. Is the manuscript technically sound, and do the data support the conclusions?

Reviewer #1: Yes

3. Has the statistical analysis been performed appropriately and rigorously? 

Reviewer #1: Yes

4. Have the authors made all data underlying the findings in their manuscript fully available?

Reviewer #1: Yes

5. Is the manuscript presented in an intelligible fashion and written in standard English?

Reviewer #1: Yes

6. Review Comments to the Author

Reviewer #1: The manuscript is now presented in a clear and coherent manner, written in Standard English, effectively conveying the importance of this work.

7. PLOS authors have the option to publish the peer review history of their article (what does this mean?). If published, this will include your full peer review and any attached files.

Reviewer #1: **Yes: **Saoussen Ben Abdallah

---

## [Editor Report · Acceptance letter]

5 Jan 2025

PONE-D-23-43180R5 

PLOS ONE

Dear Dr. Zhang, 

I'm pleased to inform you that your manuscript has been deemed suitable for publication in PLOS ONE. Congratulations! Your manuscript is now being handed over to our production team.

Kind regards, 

on behalf of

Dr. David Chau 

Academic Editor

PLOS ONE